# Visual field differences in temporal synchrony processing for audio-visual stimuli

**Yasuhiro Takeshima** *

Department of Psychology, Doshisha University, Kyotanabe-shi, Kyoto, Japan

* yasuhiro.takeshima@gmail.com

## Abstract

Audio-visual integration relies on temporal synchrony between visual and auditory inputs. However, differences in traveling and transmitting speeds between visual and auditory stimuli exist; therefore, audio-visual synchrony perception exhibits flexible functions. The processing speed of visual stimuli affects the perception of audio-visual synchrony. The present study examined the effects of visual fields, in which visual stimuli are presented, for the processing of audio-visual temporal synchrony. The point of subjective simultaneity, the temporal binding window, and the rapid recalibration effect were measured using temporal order judgment, simultaneity judgment, and stream/bounce perception, because different mechanisms of temporal processing have been suggested among these three paradigms. The results indicate that auditory stimuli should be presented earlier for visual stimuli in the central visual field than in the peripheral visual field condition in order to perceive subjective simultaneity in the temporal order judgment task conducted in this study. Meanwhile, the subjective simultaneity bandwidth was broader in the central visual field than in the peripheral visual field during the simultaneity judgment task. In the stream/bounce perception task, neither the point of subjective simultaneity nor the temporal binding window differed between the two types of visual fields. Moreover, rapid recalibration occurred in both visual fields during the simultaneity judgment tasks. However, during the temporal order judgment task and stream/bounce perception, rapid recalibration occurred only in the central visual field. These results suggest that differences in visual processing speed based on the visual field modulate the temporal processing of audio-visual stimuli. Furthermore, these three tasks, temporal order judgment, simultaneity judgment, and stream/bounce perception, each have distinct functional characteristics for audio-visual synchrony perception. Future studies are necessary to confirm the effects of compensation regarding differences in the temporal resolution of the visual field in later cortical visual pathways on visual field differences in audio-visual temporal synchrony.

## Introduction

Daily, we perceive external environments using multisensory information. Temporal synchrony is critical for the integration of different sensory stimuli. In audio-visual stimuli,

**Data Availability Statement:** All raw data and R script files are available from the OSF database (https://osf.io/2x73g/?view_only= 1dd0968b3b5d4a65a90c4ed78b1dde19).

**Funding:** YT JP20K14269 the Japan Society for the Promotion of Science https://www.jsps.go.jp/english/e-grants/index.html No

**Competing interests:** The authors have declared that no competing interests exist.

synchronously presented sounds improve visual performance. For example, visual target detectability is facilitated by simultaneous sounds in the backward masking paradigm [1]. Moreover, improvements in visual performance when accompanied by simultaneous sounds have been reported in various experimental tasks [2–4].

Temporal synchrony for audio-visual inputs is based on matching the salient temporal features of each sensory modality [5–7]. Humans perceive temporal synchrony between visual and auditory inputs, despite natural lags in arrival and processing times [8]. Thus, synchrony perception is malleable for audio-visual stimuli. For example, the point of subjective simultaneity (PSS) for light and sound, which is perceived as maximally simultaneous, differs based on viewing distance [9–11]. Moreover, a tolerant temporal binding window (TBW) has been found with regard to audio-visual simultaneity perception [12], with sounds of 100–200 ms that precede and follow [13, 14].

Temporal synchrony perception for audio-visual stimuli can be adapted to temporal lags between sensory inputs. Fujisaki, Shimojo, Kashino, and Nishida [15] reported that repeated exposure to audio-visual asynchrony shifts the PSS toward the leading stimuli, which is known as temporal recalibration. Temporal recalibration has been observed in both the temporal order judgment (TOJ) task [16] and Fujisaki et al.'s [15] simultaneity judgment (SJ) task. Moreover, van der Burg, Alais, and Cass [17] showed temporal recalibration without adaptation period, which they termed "rapid recalibration." In rapid recalibration, the PSS of the current trial was calculated to be contingent upon audio-visual asynchrony in the preceding trial. Rapid recalibration exhibits larger transient effects compared with typical recalibration with a cross-modal adaptation procedure [18].

Temporal synchrony perception for audio-visual stimuli is affected by the processing speed of the visual system. Previous studies have reported that the PSS for audio-visual stimuli is directed to more auditory leads in low spatial frequency stimuli than in high spatial frequency stimuli [19, 20]. Visual systems are composed of at least two spatio-temporal channels, namely, transient and sustained channels, each having different temporal resolutions (processing speeds) [21]. Transient channels respond to low spatial frequencies and exhibit high temporal resolution, whereas sustained channels exhibit high spatial frequencies and low temporal resolution [22]. Therefore, auditory stimuli should be presented earlier for low spatial frequency stimuli than high spatial frequency stimuli for subjective simultaneity to perceive. The retinal positions of vision are similarly different in these two channels: central vision has a low temporal resolution, while peripheral vision has a high temporal resolution [23]. Furthermore, there is a difference in visual latency between the central and peripheral visual fields (VFs). The response time for the visual stimulus reflects the difference in visual latency [22]. Therefore, the response time is shorter at the central VF than it is at the peripheral VF [24]. From the above, it can be seen that the processing speed differs between the central and peripheral VFs. The present study confirmed the effects of visual temporal resolution on audio-visual synchrony perception by manipulating the VF in which the visual stimulus was presented (i.e., eccentricity). If synchrony perception for audio-visual stimuli follows differences in temporal resolution, then the TBW of the central VF would be wider than that of the peripheral VF for tolerant synchrony perception, because the timing perception of visual presentation is ambiguous in the central VF due to low temporal resolution. However, if it follows the difference in visual latency, then the PSS of the central VF would be lower than that of the peripheral VF as previous studies manipulating spatial frequency of visual stimuli [19, 20].

The current study examines the difference in audio-visual synchrony perception between central and peripheral VFs in a multifaceted manner. Kopinska and Harris [10] compared the PSS between central (0˚ eccentricity) and peripheral (20˚ eccentricity) VFs using the TOJ task by manipulating both the eccentricity of the visual stimulus and participants' viewing distance

and showed that the timing of subjective simultaneity did not differ between VFs. However, their study did not employ other paradigms to measure PSS nor did it measure the TBW. Therefore, in this study, the PSS and TBW for audio-visual stimuli were measured using TOJ, SJ, and stream/bounce (SB) perception [25]. The underlying mechanisms differ between the TOJ and SJ tasks. The TOJ task reflects the temporal discrimination processes, whereas the SJ task reflects the temporal binding processes [26]. Furthermore, the differences between the TOJ and SJ tasks stem from their decisional and response processes [27]. Van Eijk, Kohlrausch, Juola, and van de Par [28] have shown a lack of correlation between SJ and TOJ PSS, and proposed a different type of sensitivity between TOJ and SJ for audio-visual asynchrony. Therefore, it is necessary to use both TOJ and SJ tasks to examine audio-visual temporal synchrony. Additionally, this study used SB perception as an implicit method to measure the PSS and TBW. In the SB display, two identical circles moving across each other can be perceived either to bounce off or to stream through each other. Sekuler et al. [25] reported that brief sound biases perception toward bouncing. The bouncing perception increases when a brief sound is simultaneously presented at the moment the circles coincide [29, 30]. Apparent causality among visual and auditory events as SB perception affects audio-visual synchrony perception in early multisensory integration processes [31]. The present study also compared PSS and TBW values measured using explicit (TOJ and SJ tasks) and implicit (SB perception) methods.

In addition, the differences in the effects of rapid temporal recalibration between SJ and TOJ tasks were investigated to confirm the effects of VFs on audio-visual temporal processing in this study. Roseboom [32] showed the opposite change in PSS due to rapid temporal recalibration between SJ and TOJ tasks: the PSS changed in the same direction as the stimulus onset asynchrony (SOA) of previous trials in the SJ task and in the opposite direction in the TOJ task. Therefore, this difference in the rapid temporal recalibration between SJ and TOJ tasks was also investigated. Using the SJ task, Takeshima [33] reported that normal rapid temporal recalibration occurred regardless of a difference in visual processing speed based on spatial frequency. The present study predicts that normal rapid temporal recalibration would be observed in both the central and peripheral VFs in the SJ task. In the TOJ task, as in Roseboom [32], temporal recalibration in the opposite direction to that in the SJ task is predicted to occur. This study also explores rapid temporal recalibration using SB perception, since previous studies have not examined this using the paradigm under investigation.

## Experiment 1

Experiment 1 measured and compared PSS and TBW values between the central and peripheral VFs in the TOJ task. Moreover, the process of rapid recalibration was investigated for both the VFs.

### Materials and methods

**Ethics statement.** All experiments reported in this paper were approved by the ethics committee of the Department of Psychology, Doshisha University, and were conducted in accordance with the Declaration of Helsinki.

**Participants.** Twenty volunteers (18 women and 2 men; mean age = 19.95 ± 1.81 years) participated in Experiment 1. To determine the sample size needed for this study, PANGEA (https://jakewestfall.shinyapps.io/pangea/) was used to calculate the power $(1-\beta)$ needed to detect a two-way interaction with following parameters: effect size $(d)$ = 0.45, variance of error = 0.333, variance of two-way interaction = 0.083, and the number of condition repetitions = 24. PANGEA indicated a power of 0.86 when a sample size was 16 participants.

Therefore, participants were recruited from a university lecture held at Doshisha University to meet this sample size. All participants orally reported normal or corrected-to-normal vision and normal audition. Participants were given 500 Japanese yen for their participation and provided written informed consent prior to participation.

**Apparatus.** Stimuli were generated and controlled using a custom-made program written with MATLAB (The MathWorks, Inc.), Psychtoolbox [34–36], and a laptop personal computer (MacBook Pro, Apple). The visual stimuli were displayed on a 21-inch cathode-ray tube monitor (Trinitron CPD-G520, Sony; resolution: 1024 × 768 pixels; refresh rate: 100 Hz). The auditory stimuli were conveyed using an audio interface (Clarett 2Pre, Focusrite) and headphones (MDR-CD900ST, Sony). The simultaneity of the visual and auditory stimuli was confirmed using a digital oscilloscope (DS-5424A, Iwatsu). The experiment was conducted in a darkened room with background noise levels of 39.8 dB (A). Participants viewed the monitor binocularly at a distance of 70 cm with their heads stabilized on a chin rest.

**Stimuli.** A white (86.64 cd/m$^2$) fixation cross (0.5 × 0.5˚) and circle (visual stimulus: 1.5˚ diameter) was presented on a black (0 cd/m$^2$) background. The fixation cross was presented at the center of the screen. The circle was presented at one of the following three locations: the center of the screen, 10˚ left, and right of the fixation stimulus. The duration of the target presentation was 10 ms. The auditory stimuli consisted of pure tones with a frequency of 500 Hz, a sound pressure level of 55 dB (A), and a duration of 10 ms (including ramp times of 2 ms at the start and end of the sound wave envelope). The onset times of the visual and auditory stimuli were synchronized. The following 10 SOAs were established between the visual and auditory stimuli: ±510, ±260, ±130, ±50, and ±10 ms (negative SOAs indicate that the auditory stimulus was presented prior to the visual stimulus, and vice versa).

**Procedure.** A schematic of the trial design is presented in Fig 1. The trials were initiated by pressing the 0 key. Each trial consisted of a 500 ms fixation period followed by blank and target displays. The duration of this blank display was a fixed length of 500 ms, plus an additional SOA when the auditory stimulus was presented prior to the visual stimulus (i.e., 500 ms at the shortest and 1010 ms at the longest). During the target display period, a circle was presented at one of the following three locations: center, left, or right. The tone either preceded or followed the onset of the circle using the SOA that had been drawn randomly from the set. After the target display, a blank display was again presented. The duration of this blank display was a fixed length of 200 ms, plus an additional SOA when the visual stimulus was presented prior to the auditory stimulus (i.e., 200 ms at the shortest and 710 ms at the longest). Then,

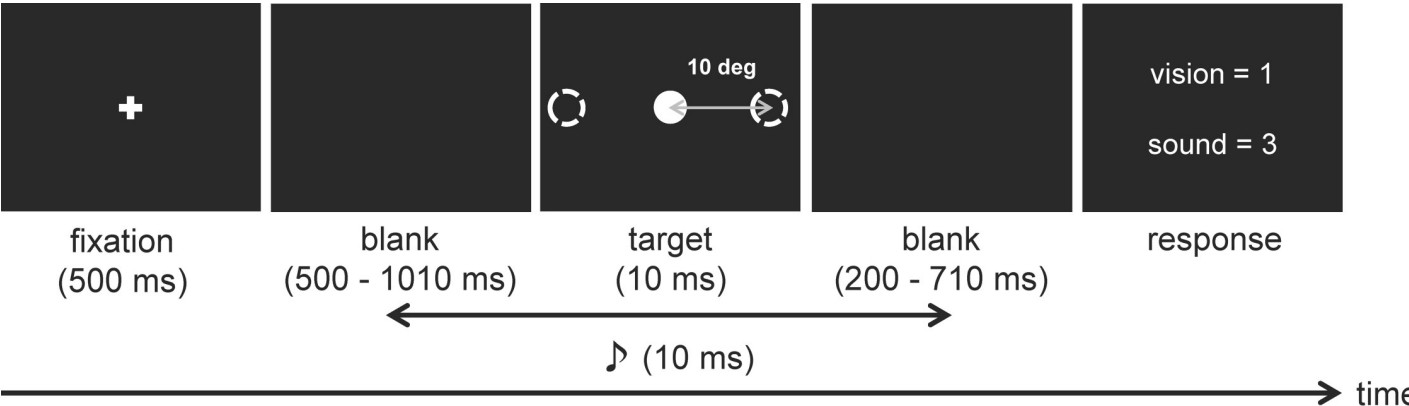

**Fig 1. The schematic representation of the procedure used in the temporal order judgment task.**

participants were instructed to judge the temporal order by pressing the 1 key for vision leads and the 3 key for audition leads. First, the participants performed 30 practice trials. The participants then completed 480 trials that were divided into six blocks. One block consisted of 80 trials, with eight trials for each SOA condition. Half the trials of one block had visual stimuli presented at the central VF, while the remaining half had visual stimuli presented at the peripheral VF (i.e., 24 repetitions for each experimental condition). Additionally, in the peripheral VF condition, the visual stimulus was presented to the left or right of the fixation stimulus with equal frequency. Participants took short breaks between the blocks.

## Results

The proportion of vision lead responses was calculated for each condition. To compute the PSS and sigma (i.e., TBW), the cumulative Gaussian function was fitted to each participant's data using the maximum-likelihood method:

$$P\left(response|SOA\right) = \frac{1}{1 + e^{\left[-\frac{1}{sigma}(SOA - PSS)\right]}}$$

The SOA parameters matched those of the experimental conditions (from -510 to +510 ms). The PSS and sigma parameters were evaluated using estimations. The sigma value was restricted to values greater than 0. One participant was excluded from further analysis, because their computed sigma value was large ($>$ 600). The results of the analysis are shown in Fig 2A, which demonstrates the mean percentages of vision lead responses as a function of VF and SOA with fitted psychometric functions (center: Mean RMSE = 0.10 ± 0.03, periphery: Mean RMSE = 0.09 ± 0.03). The PSS and sigma results are shown in Fig 2B and 2C. The PSS was smaller in the center condition than in the periphery condition, $t(18) = 2.33$, $p < .05$, $d = 0.51$, whereas sigma did not differ between the center and periphery conditions, $t(18) = 1.09$, $p = .29$, $d = 0.13$.

An inter-trial analysis was conducted to examine whether the modality order in a given previous trial (Trial t-1) affected the distribution of vision leads responses in the current trial (Trial t). The VFs in the previous trial (Trial t-1) were not split because the correspondence of the spatial location between the current (Trial t) and previous (Trial t-1) trials did not affect the rapid recalibration [37]. Further, the distribution of perceived vision leads as a function of

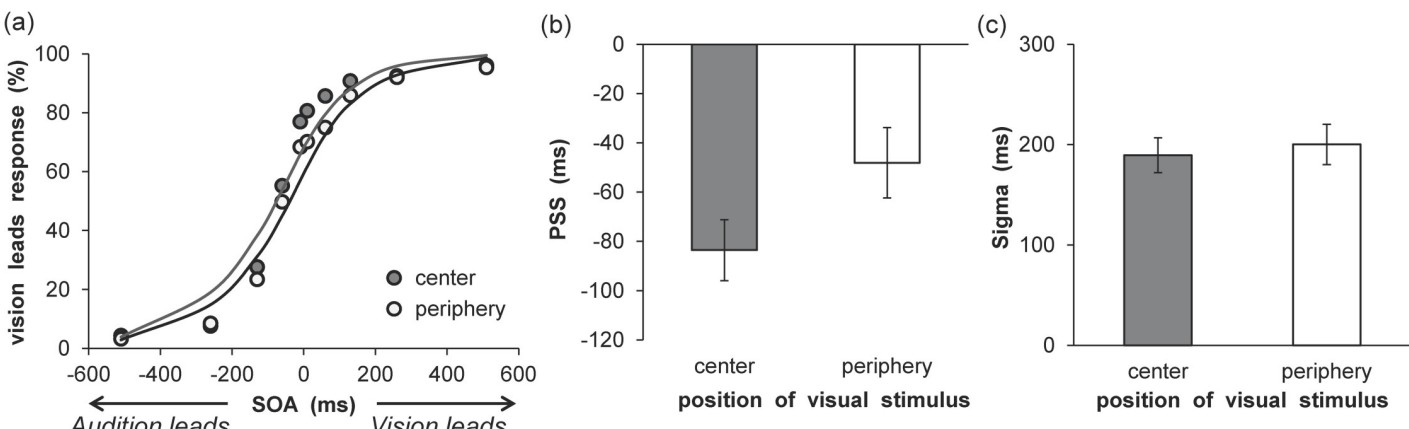

**Fig 2. Results of synchrony perception analyses in Experiment 1.** (a) Mean percentage of vision lead responses. The average fitting functions are plotted for the central and peripheral visual field data. (b) Mean estimated point of subjective simultaneity. (c) Mean estimated sigma. Error bars represent standard errors of the mean ($n = 19$).

SOA was compiled for each participant and each VF separately, given the cases in which trial t-1 exhibited either a negative SOA (i.e., audition leads) or positive SOA (i.e., vision leads). The total distributions were then fitted to a cumulative Gaussian function. The results are shown in Fig 3A and 3B, which demonstrate the mean percentages of vision lead responses as a function of VF and SOA with fitted psychometric functions for both modality orders (center/audition leads: Mean RMSE = 0.12 ± 0.04, center/vision leads: Mean RMSE = 0.10 ± 0.05, periphery/audition leads: Mean RMSE = 0.11 ± 0.03, periphery/vision leads: Mean RMSE = 0.11 ± 0.04). The recalibration shifts are summarized in Fig 3C, which plots the PSS for both modality orders and the two different VFs. For the PSS, a two-way analysis of variance (ANOVA) with modality order (2) and VF (2) was conducted. The results revealed a significant main effect of VF, $F(1, 18) = 5.55$, $p < .05$, $\eta_p^2 = .24$, and a two-way interaction, $F(1, 18) = 14.14$, $p < .01$, $\eta_p^2 = .44$. The simple main effect of the modality order was significant at the

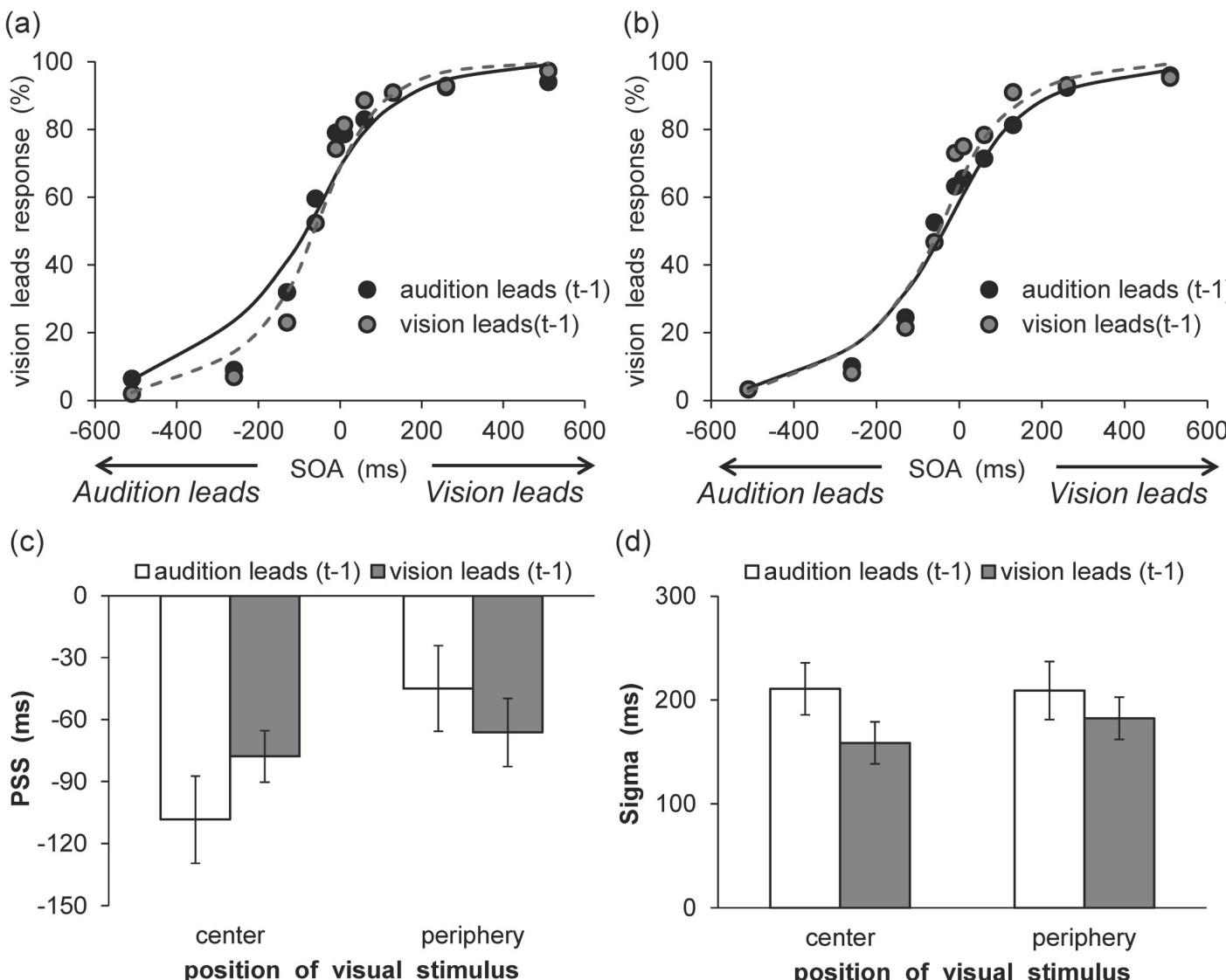

**Fig 3. Results of rapid recalibration analyses in Experiment 1.** Mean percentage of vision lead responses in the (a) central and (b) peripheral visual fields. The average fitting functions are plotted for preceding trials of audition leads and vision leads data. (c) Mean estimated point of subjective simultaneity. (d) Mean estimated sigma. Error bars represent standard errors of the mean ($n = 19$).

central VF, $F(1, 18) = 4.94$, $p < .05$, $\eta_p^2 = .22$, which indicates that the PSS was lower in the audition leads condition than in the vision leads condition. However, this simple main effect was not significant for the peripheral VF, $F(1, 18) = 2.74$, $p = .12$, $\eta_p^2 = .13$. Moreover, the simple main effect of VF was significant in the audition leads condition, $F(1, 18) = 12.71$, $p < .01$, $\eta_p^2 = .41$, which indicates that the PSS was lower in the center than in the periphery. However, this simple main effect was not significant in the vision leads condition, $F(1, 18) = 0.48$, $p = .50$, $\eta_p^2 = .03$. The main effect of modality order was not significant, $F(1, 18) = 0.13$, $p = .77$, $\eta_p^2 = .01$. The sigma values for both modality order and the two different VFs are shown in Fig 3D. For sigma, a two-way ANOVA with modality order (2) and VF (2) was conducted. The results revealed a significant main effect of modality order, $F(1, 18) = 11.91$, $p < .01$, $\eta_p^2 = .40$, which indicates that the sigma of the vision leads was lower than that of the audition leads. However, the main effect of VF, $F(1, 18) = 1.15$, $p = .30$, $\eta_p^2 = .06$, and the two-way interaction, $F(1, 18) = 2.87$, $p = .11$, $\eta_p^2 = .14$, were not significant.

## Discussion

Experiment 1 compared the synchrony perception and rapid recalibration processes between VFs using the TOJ task. The PSS was smaller at the center than at the periphery, whereas the TBW did not differ between VFs. Moreover, normal rapid recalibration occurred only in the central VF, but not in the peripheral VF. Therefore, presentation of the vision leads to a preceding stimulus that causes its TBW to become narrower than that of the audition leads.

The difference in the PSS observed between the central and peripheral VFs was consistent with the difference in visual latency between the central and peripheral VFs. The response time for the visual stimulus was shorter in the central VF than in the peripheral VF [24]. Previous studies that manipulated spatial frequency also observed differences in the PSS consistent with the differences observed in response time [19, 29]. The difference in visual latency is reflected in the response time of the visual stimulus [22]. Therefore, in a TOJ task, differences in the PSS between the central and peripheral VFs could be attributed to differences in visual latency rather than temporal resolution.

The results of normal rapid recalibration for the central VF were inconsistent with the predictions based on previous studies. Roseboom [32] showed that the PSSs shifted in opposite directions from normal rapid temporal recalibration in a TOJ task. Moreover, Keane, Bland, Matthews, Carroll, and Wallis [38] found that opposite-directed PSS shifts were induced by choice-repetition bias in a TOJ task. Choice-repetition bias refers to the tendency to repeat judgments of the temporal order of a previous trial in a current trial. Additionally, rapid recalibration was obfuscated by opposite-directed PSS shifts to a choice-repetition bias [38]. It is possible that the choice-repetition bias was suppressed in the central VF in this study. Low information reliability induces a larger choice-repetition bias [39, 40]. Therefore, the reliability of judging temporal order for audio-visual stimuli would be high in the central VF. This speculation needs to be further investigated.

Moreover, the difference in the TBW between audition and vision leads during the rapid recalibration process is a novel finding. In Experiment 1, the TBW width narrowed in the preceding vision leads presentation than in the preceding audition leads presentation. A narrow TBW indicates high sensitivity for judging the temporal order between visual and auditory stimuli. Therefore, this finding shows that the temporal information of visual precedence in a previous trial increases sensitivity in a TOJ task for audio-visual stimuli. In previous studies of rapid recalibration (e.g., [17, 28, 36]), the difference in TBW width between the audition and vision leads conditions has not been investigated. Such a difference in the TBW was not predicted in this study, and this needs to be examined in more detail in the future.

## Experiment 2

Experiment 2 measured and compared PSS and TBW between the central and peripheral VF in the SJ task. Moreover, the process of rapid recalibration was investigated in both VFs to provide robust evidence.

### Materials and methods

**Participants.**   Twenty-one volunteers (14 women and 7 men; mean age = 21.57 ± 2.17 years) participated in Experiment 2. The required sample size was determined based on the same criterion as in Experiment 1, and participants were recruited to meet this sample size from a university lecture held at Doshisha University. All participants orally reported normal or corrected-to-normal vision and normal audition. Participants were compensated with 500 Japanese yen for their participation and provided written informed consent prior to participation.

**Stimuli.**   The same visual and auditory stimuli as in Experiment 1 were used. The onset times of the visual and auditory stimuli were synchronized. The following nine SOAs were present between the visual and auditory stimuli: ±510, ±260, ±130, ±50, and 0 ms (negative SOAs indicate that the auditory stimulus was presented prior to the visual stimulus, and vice versa).

**Procedure.**   The trial sequence was nearly the same as that of Experiment 1, except for the response period. During the response period, participants were instructed to press the 1 key for simultaneity and the 3 key for asynchrony. First, the participants performed 30 practice trials. The participants then completed 480 trials that were divided into six blocks. One block comprised 80 trials, with eight trials for each SOA condition (i.e., 16 trials for 0 ms SOA and eight trials for other SOAs). For half of the trials of one block, visual stimuli were presented in the central VF, and for the other half, visual stimuli were presented at the peripheral VF. Additionally, in the peripheral VF condition, visual stimuli were presented to the left or right of the fixation stimulus with equal frequency. Participants took short breaks between the blocks.

### Results

The proportion of simultaneity responses was calculated for each condition. To compute the alpha, PSS, and sigma values, the three-parameter Gaussian function was fitted to each participant's data per minimization of the root-mean-square error (RMSE):

$$\text{P}\left(response|SOA\right) = Alpha \cdot e^{\left[-.5\left(\frac{SOA-PSS}{Sigma}\right)^2\right]}$$

The SOA parameter was equal to that of the experimental conditions (from -510 to +510 ms). The alpha, PSS, and sigma parameters were estimated, and these parameters indicated the height, peak position on the SOA axis, and the width of the Gaussian function, respectively. The sigma value was restricted to greater than 0, and the alpha value was restricted between 0 and 1. One participant was excluded from further analysis because their computed sigma value was large ($>$ 600). The results are shown in Fig 4A, which presents the mean percentages of simultaneity responses, as a function of VF and SOA with fitted psychometric functions (center: Mean RMSE = 0.07 ± 0.03, periphery: Mean RMSE = 0.07 ± 0.03). Moreover, PSS and sigma are shown in Fig 4B and 4C. The PSS value did not differ between the center and periphery conditions, $t$ (19) = 0.22, $p$ = .83, $d$ = 0.02, whereas the sigma value was larger in the center condition than in the periphery condition, $t$ (19) = 4.99, $p <$ .001, $d$ = 0.41. A difference in alpha value was also not observed between the center ($M$ = 0.99 ± 0.03) and periphery ($M$ = 0.97 ± 0.06) conditions, $t$ (19) = 1.50, $p$ = .15, $d$ = 0.40.

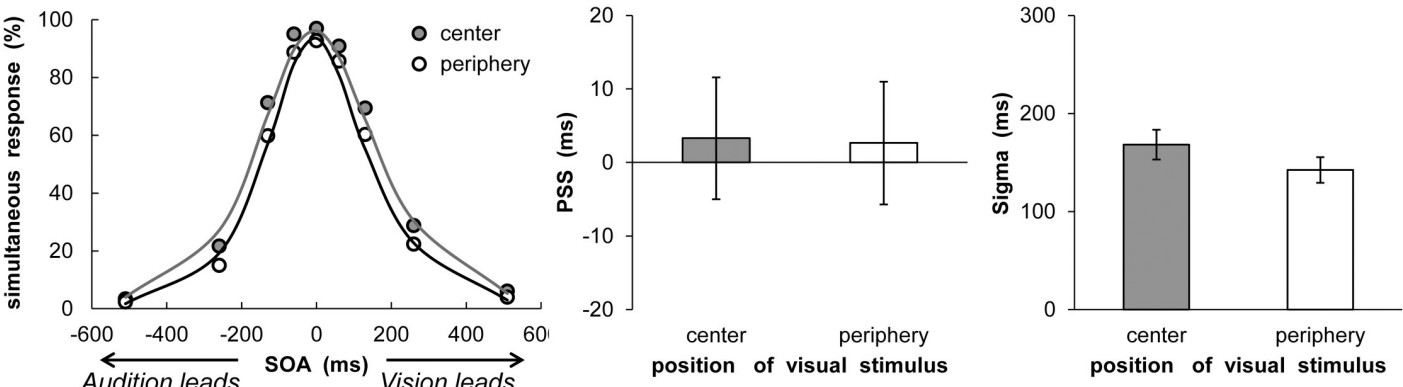

**Fig 4. Results of synchrony perception analyses in Experiment 2.** (a) Mean percentage of simultaneous responses. The average fitting functions are plotted for the central and peripheral visual field data. (b) Mean estimated point of subjective simultaneity. (c) Mean estimated sigma. Error bars represent standard errors of the mean ($n = 20$).

The same inter-trial analysis was conducted as in Experiment 1. Distributions of perceived simultaneity as a function of SOA were compiled for each participant and each VF separately for cases in which trial t-1 exhibited a negative SOA (i.e., audition leads) or positive SOA (i.e., vision leads). These distributions were subsequently fitted using the Gaussian function described above. The results are shown in Fig 5A and 5B, which represent the mean percentages of simultaneity responses, as a function of VF and SOA with fitted psychometric functions for both modality order (center/audition leads: Mean RMSE = 0.10 ± 0.03, center/vision leads: Mean RMSE = 0.10 ± 0.03, periphery/audition leads: Mean RMSE = 0.09 ± 0.03, periphery/vision leads: Mean RMSE = 0.09 ± 0.04). The recalibration shifts are summarized in Fig 5C, which plots the PSS for both modality order and the two different VFs. For the PSS, a two-way ANOVA with modality order (2) and VF (2) was conducted. The results revealed a significant main effect of modality order, $F(1, 19) = 14.91$, $p < .01$, $\eta_p^2 = .44$, which indicates that the PSS was larger for visual leads than for audition leads. However, the main effect of VF, $F(1, 19) = 0.80$, $p = .38$, $\eta_p^2 = .04$, and the two-way interaction, $F(1, 19) = 0.76$, $p = .39$, $\eta_p^2 = .04$, were not significant. Furthermore, the sigma values for both modality orders and the two different VFs are shown in Fig 5D. For sigma, a two-way ANOVA with modality orders (2) and VF (2) was conducted. The results reveal a significant main effect of VF, $F(1, 19) = 27.01$, $p < .001$, $\eta_p^2 = .59$, which indicates that the sigma values of the center condition were larger than those of the periphery condition. However, the main effect of modality order, $F(1, 19) = 0.70$, $p = .41$, $\eta_p^2 = .04$, and the two-way interaction, $F(1, 19) = 0.03$, $p = .86$, $\eta_p^2 = .002$, were not significant.

## Discussion

Experiment 2 compared the synchrony perception and rapid recalibration process between VFs using the SJ task. The TBW was wider at the center than at the periphery whereas the PSS was nearly identical between the VFs. Rapid recalibration occurred under both VF conditions. However, the magnitude of the recalibration did not differ between the center and the periphery.

In Experiment 2, a VF difference in the TBW consistent with the difference in temporal resolution, was observed. The central VF projects to the sustained channel, whereas the peripheral VF projects to the transient channel [23]. The temporal resolution of the sustained channel is lower than that of the transient channel [22]. Thus, the range of audio-visual synchrony

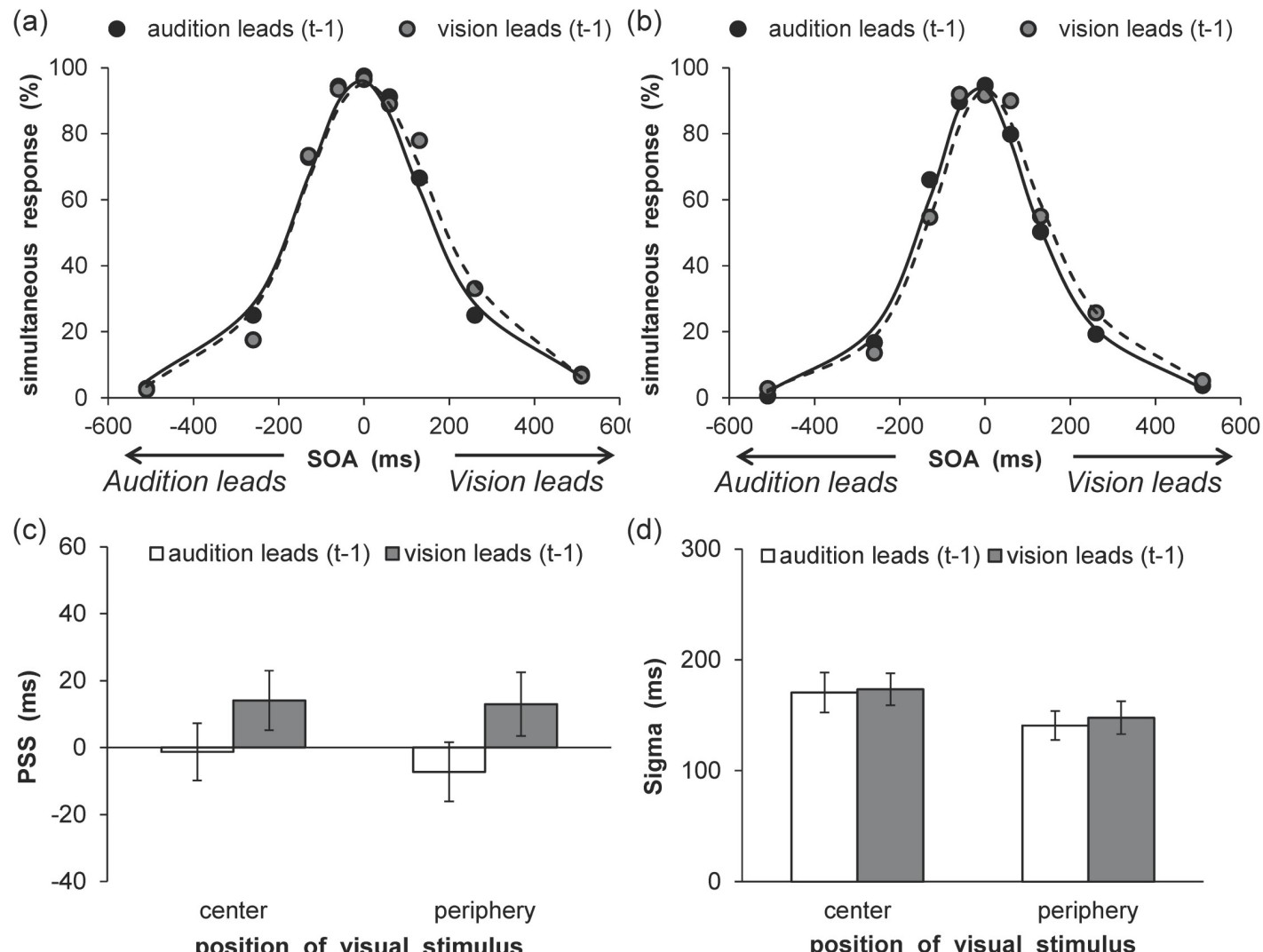

**Fig 5. Results of rapid recalibration analyses in Experiment 2.** Mean percentage of simultaneous responses in the (a) central and (b) peripheral visual fields. The average fitting functions are plotted for preceding trials of audition leads and vision leads data. (c) Mean estimated point of subjective simultaneity. (d) Mean estimated sigma. Error bars represent standard errors of the mean ($n = 20$).

perception would be more tolerant in the central VF due to adapting to low temporal resolution. Stevenson, Kruger Fister, Barnett, Nidiffer, and Wallace [40] have found a broader TBW in the peripheral VF than in the central VF. However, a higher percentage of simultaneous judgment was observed in the peripheral VF outside of the present experiment in their study (60˚ and 90˚ eccentricities). In the peripheral VF (30˚ eccentricity), which was closer to the present experiment, the percentage of simultaneous judgment was slightly lower than that in the central VF. Therefore, the TBW of the central VF is assumed to become wider than that of the peripheral VF up to a certain range (at least 30˚ eccentricity), which is attributed to the temporal resolution.

However, the difference in temporal resolution due to VFs did not affect the magnitude of rapid recalibration. Furthermore, unlike Experiment 1, the rapid recalibration process occurred in both VFs. The differences in the temporal resolution did not modulate the intensity of the rapid recalibration process in the SJ task.

## Experiment 3

In Experiments 1 and 2, the PSS and TBW were measured using explicit methods (i.e., TOJ and SJ tasks). In Experiment 3, the PSS and TBW were measured using an implicit method using SB perception. Moreover, the process of rapid recalibration was investigated in both VFs using this method.

### Materials and methods

**Participants.** Twenty-one volunteers (14 women and 7 men; mean age = 23.19 ± 2.84 years) participated in Experiment 3. The required sample size was determined based on the same criterion as in Experiment 1, and participants were recruited to meet this sample size from a university lecture held at Doshisha University. All participants orally reported normal or corrected-to-normal vision and normal audition. Participants received 500 Japanese yen for their participation, and their written informed consent was obtained prior to participation.

**Stimuli.** Two white disks (1.5˚ in diameter) appeared 3.0˚ above the fixation stimulus and were initially separated by 4.5˚. The two disks in the central VF condition moved laterally toward one another, coincided, and continued moving until they were 2.25˚ from the point of superposition at a speed of 3.13˚/s. In the peripheral VF condition, the superposition was 10˚ left or right from the superposition of the center condition. The auditory stimulus was the same as in Experiments 1 and 2. The following nine SOAs were present between the visual and auditory stimuli: ±510, ±260, ±130, ±50, and 0 ms (negative SOAs indicate that the auditory stimulus was presented prior to the visual stimulus, and vice versa). In the 0 ms SOA condition, the auditory stimulus was presented at the instance of superposition.

**Procedure.** The trials were initiated by pressing the 0 key. Each trial consisted of a 500 ms fixation stimulus followed by motion displays (for a duration of 1440 ms, see Fig 6). The tone either preceded or followed the instance of superposition by the SOA that was drawn randomly from the set. During the motion display period, participants were instructed to gaze the fixation cross. After the motion display, participants were instructed to judge the motion trajectory by pressing the 1 key for bounce and the 3 key for stream. The participants completed 480 trials divided into six blocks. One block comprised 80 trials, with eight trials for each SOA condition (i.e., 16 trials for 0 ms SOA and eight trials for other SOAs). For half of the trials of one block, visual stimuli were presented at the central VF, and for the other half, visual stimuli were presented at the peripheral VF. Additionally, for the peripheral VF condition, visual stimuli were presented at the left or right of the fixation stimulus with equal frequency.

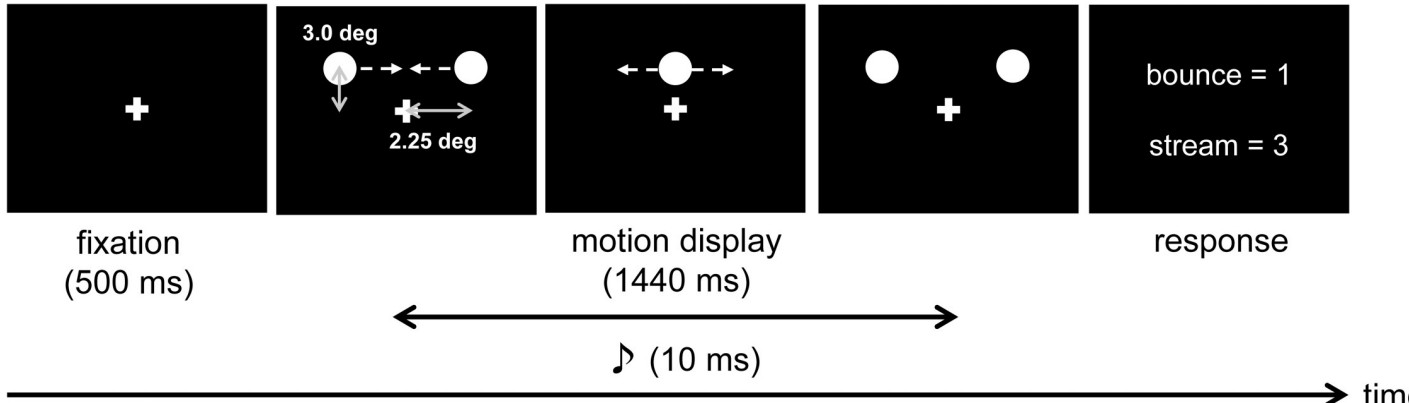

**Fig 6. The schematic representation of the procedure used in the stream/bounce display.**

Participants took short breaks between the blocks. Prior to performing this main task, participants performed 30 practice trials, and then judged the same motion trajectory without sound in 48 trials (24 trials each for the central and peripheral conditions).

## Results

The proportion of bounce responses was calculated for each condition. At first, the bounce responses were significantly larger at the center ($M$ = 38.54%, $SEM$ = 5.39) than at the periphery ($M$ = 27.08%, $SEM$ = 4.15) without sound, $t$ (16) = 2.18, $p < .05$, $d = 0.49$. To compute the alpha, PSS, and sigma values, a three-parameter Gaussian function was fitted to each participant's data to minimize the RMSE, as in Experiment 2. The data of four participants were excluded from further analysis because the computed sigma values were large ($> 600$: two participants), and the bounce responses were less than 5% in all conditions (two participants). The results are shown in Fig 7A, which represents the mean percentages of bounce responses, as a function of the VF and SOA with fitted psychometric functions (center: Mean RMSE = 0.08 ± 0.03, periphery: Mean RMSE = 0.07 ± 0.01). Moreover, the PSS and sigma values are shown in Fig 7B and 7C. The PSS values did not differ between the center and periphery conditions, $t$ (16) = 0.41, $p = .69$, $d = 0.06$. Moreover, the sigma value did not differ between the center and periphery conditions, $t$ (16) = 0.83, $p = .42$, $d = 0.16$. The alpha value was larger in the center ($M$ = 0.82 ± 0.14) than in the periphery ($M$ = 0.74 ± 0.19) condition, $t$ (16) = 2.57, $p < .05$, $d = 0.47$.

The same inter-trial analysis of Experiments 1 and 2 was conducted. Distributions of perceived bounce responses as a function of SOA were compiled for each participant and each VF separately for cases in which trial t-1 exhibited a negative SOA (i.e., audition leads) or positive SOA (i.e., vision leads). These distributions were subsequently fitted with the same Gaussian function as in Experiment 2. The results are shown in Fig 8A and 8B, which represent the mean percentages of bounce responses as a function of VF and SOA with fitted psychometric functions for both modality order (center/audition leads: Mean RMSE = 0.10 ± 0.04, center/vision leads: Mean RMSE = 0.10 ± 0.04, periphery/audition leads: Mean RMSE = 0.09 ± 0.04, periphery/vision leads: Mean RMSE = 0.10 ± 0.03). The recalibration shifts are summarized in Fig 8C, which plots the PSS for both modality order and the two different VFs. For the PSS, a two-way ANOVA with modality order (2) and VF (2) was conducted. The results revealed a marginally significant main effect of modality order, $F$ (1, 16) = 4.16, $p = .06$, $\eta_p^2 = .21$.

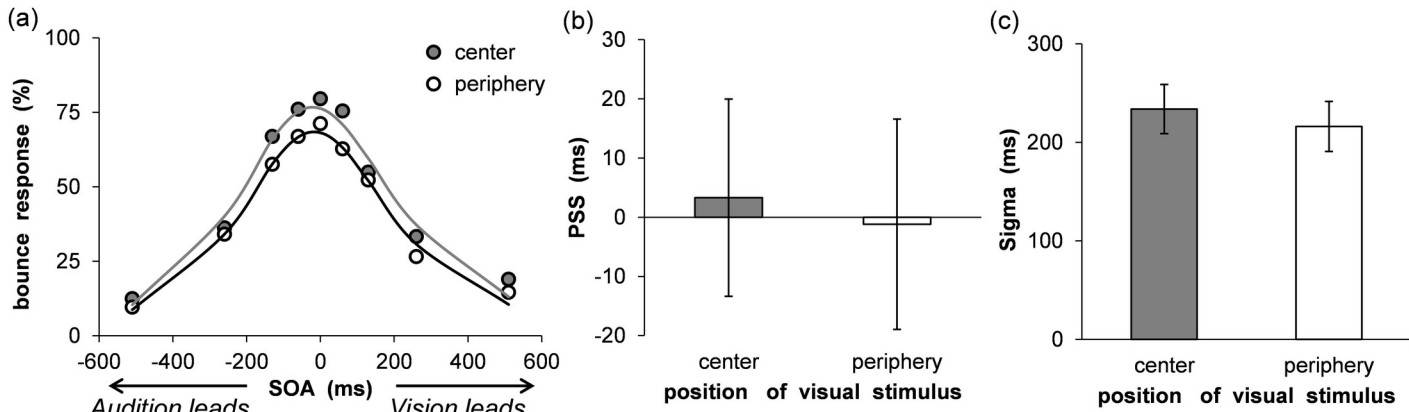

**Fig 7. Results of synchrony perception analyses in Experiment 3.** (a) Mean percentage of bounce responses. The average fitting functions are plotted for the central and peripheral visual field data. (b) Mean estimated point of subjective synchrony. (c) Mean estimated sigma. Error bars represent standard errors of the mean ($n$ = 17).

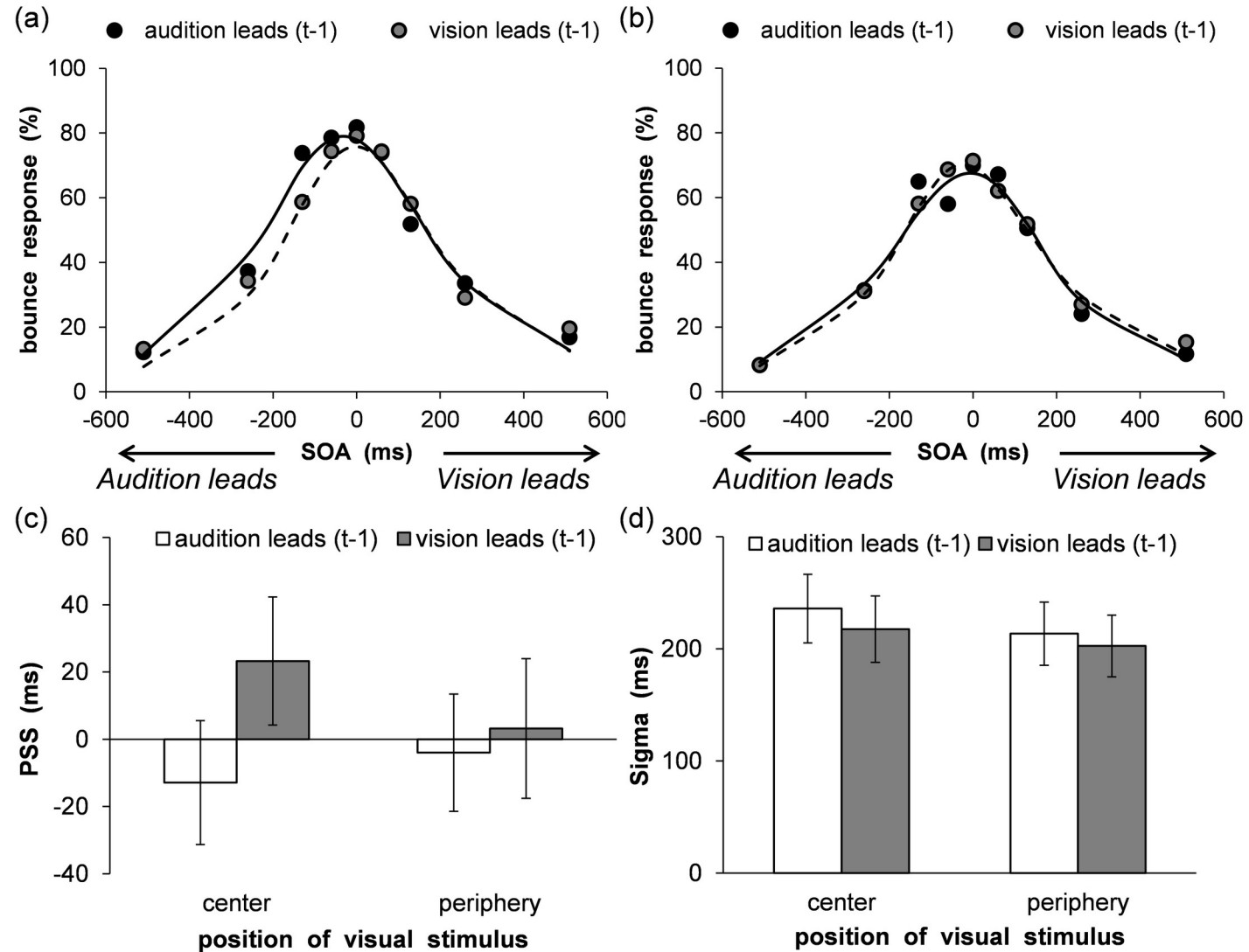

**Fig 8. Results of rapid recalibration analyses in Experiment 3.** Mean percentage of bounce responses in the (a) central and (b) peripheral visual fields. The average fitting functions are plotted for preceding trials of audition leads and vision leads data. (c) Mean estimated point of subjective simultaneity. (d) Mean estimated sigma. Error bars represent standard errors of the mean ($n = 17$).

However, the main effect of VF, $F(1, 16) = 0.26$, $p = .62$, $\eta_p^2 = .02$, and the two-way interaction, $F(1, 16) = 2.02$, $p = .17$, $\eta_p^2 = .11$, were not significant. To confirm the occurrence of rapid temporal recalibration, a one-sample $t$-test (two-tailed) was conducted for the magnitude of rapid recalibration (calculated by subtracting PSS with auditory leads (t-1) from one with vision leads (t-1)) in both VFs. The results indicated that a significant difference was observed only in the center, $t(16) = 2.39$, $p < .05$, $d = 0.58$, but not in the periphery, $t(16) = 0.50$, $p = .62$, $d = 0.12$. Furthermore, the sigma values for both modality orders and the two different VFs are shown in Fig 8D. For sigma, a two-way ANOVA with modality orders (2) and VF (2) was conducted. The results did not reveal a significant main effect of VF, $F(1, 16) = 0.76$, $p = .40$, $\eta_p^2 = .05$, modality order, $F(1, 16) = 0.81$, $p = .38$, $\eta_p^2 = .05$, and the two-way interaction, $F(1, 16) = 0.06$, $p = .81$, $\eta_p^2 = .004$.

## Discussion

Experiment 3 compared synchrony perception and rapid recalibration process between VFs using SB perception. Both the PSS and TBW were nearly identical to the physical synchrony (i.e., 0 ms) in both VFs. Moreover, rapid recalibration occurred only in the central VF, but not in the peripheral VF.

Audio-visual temporal processing was not affected by VF differences in SB perception. SB perception did not differ between the central and peripheral VFs in terms of the PSS and TBW. However, the duration of motion display was over 1000 ms and participants' eye movements were not monitored in this experiment. Thus, the participants' gaze was off the fixation point in the motion display period, which may have affected the current results.

The discrepancy in the results between the central and peripheral VF in rapid recalibration could be explained by this difference in judgment. In this experiment, normal rapid recalibration occurred only in the central VF and not in the peripheral VF. It is possible that participants performed timing judgment of bounce in the central VF and causal judgment of bounce in the peripheral VF. The occurrence of rapid recalibration due to SB perception has not yet been examined. If participants performed a causal judgment in peripheral VF and the timing information on a previous trial did not cause a change in the PSS in causal judgment, it is consistent with the results of this experiment. Future studies are needed to confirm this hypothesis, as there are few experimental data on rapid recalibration using SB perception.

## General discussion

The present study examined the different characteristics of temporal synchrony perception for audio-visual stimuli based on the VF. In this study, the following three types of measurements were used: TOJ, SJ, and SB perception tasks. Experiment 1 showed that the PSS value was smaller in the central VF condition than in the peripheral VF condition. Moreover, rapid recalibration did not occur in peripheral VF conditions in Experiment 1. Experiment 2 indicated that the TBW was broader at the central VF than at the peripheral VF. However, the rapid recalibration magnitude did not differ between the central and peripheral VFs in Experiment 2. In Experiment 3, neither the PSS nor the TBW differed between the central and peripheral VFs. Moreover, rapid recalibration was observed only in the central VF (see Table 1).

VF differences in audio-visual temporal synchrony were observed in the TOJ and SJ tasks in this study. In the TOJ task of Experiment 1, the PSS score indicated that the auditory stimulus was presented earlier as a visual stimulus for one to perceive subjective simultaneity in the central than in the peripheral VF condition. In the SJ task of Experiment 2, the TBW width of the central VF was broader than that of the peripheral VF. The difference in the PSS of the TOJ task was consistent with the VF difference in visual latency, whereas the difference in the

**Table 1. Summary of the three experiments' main findings.**

| Exp. | Synchrony perception | | Rapid recalibration | |
|---|---|---|---|---|
| | PSS | TBW | Cent. | Peri. |
| Exp. 1 (TOJ) | Cent. < Peri. | Cent.≒Peri. | A. leads < V. leads | A. leads≒V. leads |
| Exp. 2 (SJ) | Cent.≒Peri. | Cent. > Peri. | A. leads < V. leads | A. leads < V. leads |
| Exp. 3 (SB) | Cent.≒Peri. | Cent.≒Peri. | A. leads < V. leads | A. leads≒V. leads |

Note. Cent. = central visual field, Peri. = peripheral visual field

A. leads = audition leads (t-1), V. leads = vision leads (t-1).

TBW of the SJ task was consistent with the VF difference in temporal resolution. As a preliminary prediction, the VF differences in the PSS and TBW both follow differences in each visual latency and temporal resolution: the PSS score was lower and the TBW width was wider in the central VF than in the peripheral VF. It has been shown that the difference in eccentricity-dependent temporal resolution is observed in the early visual cortex and is compensated later in the cortical visual pathway [41]. TOJ tasks are assumed to be associated with higher-order processing compared to SJ tasks [28, 42]. Therefore, what PSS score was affected by visual latency could be attributed to the visual hierarchy of eccentricity-dependent temporal contrast in a TOJ task. Moreover, TOJ tasks are proposed to be sensitive to PSS changes [28], whereas an SJ task is proposed as sensitive to TBW change [43]. Thus, the discrepancy in the indices for which VF differences were observed between the TOJ and SJ tasks would reflect the difference in sensitivity of indices to audio-visual synchrony perception.

The present study also revealed PSS differences between the TOJ and SJ tasks. The TOJ task showed negative PSS values (i.e., participants interpreted them as synchronous when the auditory stimulus led the visual stimulus for the pair), whereas the SJ task indicated positive PSS values (i.e., participants interpreted as synchronous when the auditory stimulus led the visual stimulus for the pair). Normally, positive PSSs have been observed in many studies [44] according to a neural delay (visual stimuli need to be presented before sound to compensate for a slow neural processing compared to auditory stimuli) and a tuning toward the natural situation (light reaches the sense organs before sounds do). In contrast, negative PSSs are obtained more from TOJ data than from SJ data [28]. The process of additional cognitive operations to label the judgment of whether auditory leads or visual leads [39] might be associated with this bias of PSS in a TOJ task. In this study, the PSS values replicated these characteristics in audio-visual synchrony perception. A model-based analysis involving the parameters to estimate the timing decisional, and response process is proposed and suggests the underlying common timing processes in TOJ and SJ tasks [45]. The VF differences for audio-visual synchrony perception also need to be re-analyzed using this approach.

Both PSS and TBW obtained from SB perception data did not reveal VF differences unlike the TOJ and SJ tasks. Because the bouncing perception is sensitive to the lag between sound onset and moment the circles coincide in the SB display [29, 30], SB perception implicitly measured audio-visual temporal processing. Fujisaki et al. [15] also measured PSS using SB perception. However, participants perceived the causality for motion trajectory and sound in an SB display, whereas simultaneity for auditory and visual stimuli in an SJ task. Discrepancies between SB and SJ have been reported [46, 47].

A VF difference in rapid recalibration was observed in the TOJ task and SB perception in the current study. In these paradigms, rapid recalibration occurred only in the central VF. However, this VF difference was not attributed to the difference in processing speed based on retinal position in either paradigm. In the TOJ task, it is assumed that choice-repetition bias, which cancels a PSS shift by rapid recalibration, did not occur. In SB perception, it is possible that the information judged was different between the central and peripheral VFs. Therefore, rapid recalibration may be induced by intersensory temporal processes that are not modulated by the visual processing speed [33].

A limitation of the present study is that the effects of the cortical magnification factor were not investigated. The areas projected by the retina are larger in the central VF than in the peripheral VF [48], and visual acuity is usually higher in the central VF than in the peripheral VF. Therefore, the size of the visual stimuli must be large to match the areas on the visual striate cortex in the peripheral VF compared to the central VF. The differences in cortical area size projected by the retina may induce differences in the visual stimulus intensity. Low-

intensity stimuli are susceptible to judge synchrony [49]. Thus, future studies should control for the cortical magnification factor.

The present study found VF differences in the temporal synchrony perception of audio-visual stimuli. These findings support the notion that differences in visual processing speed modulate temporal processing in audio-visual integration [19]. However, these VF differences could not be explained simply by the difference in the temporal resolution of the VF. The compensation of temporal resolution difference in the late cortical visual pathway [42] may be associated, and future studies are necessary to support this speculation. Furthermore, a paradigm difference in temporal synchrony perception was also observed in this study, among TOJ, SJ, and SB. The PSS score tends to be a negative value (i.e., interpreted as synchrony when auditory stimulus leads to the visual stimulus) in a TOJ task compared to an SJ task [28]. SB perception is also likely to show a negative PSS value compared to an SJ task, since causality judgment is performed in SB perception [47]. In this study, the PSS scores indicated a similar tendency to these functional characteristics. One of the potential contributions is that each of these three methods measured different functional characteristics for audio-visual synchrony perception in a series of experiments. Additionally, the present findings suggest that explicit measurements (i.e., TOJ and SJ tasks) are more desirable for examining the temporal synchrony perception of audio-visual stimuli between VFs, because of the different judgments that occur when using an implicit measurement (i.e., SB perception).

## Conclusion

This study demonstrates that temporal synchrony processing for audio-visual stimuli is modulated by differences in processing speed between the central and peripheral VFs. These functions differed based on the experimental tasks involved, in which the PSS value became smaller in the central VF in the TOJ task, and the TBW became broader in the central VF in the SJ task. In addition, rapid temporal recalibration occurred only at the central VF in the TOJ task and SB perception, whereas at both the central and peripheral VF in the SJ task. These inconsistent results could be associated with the compensation of temporal resolution differences in the late cortical visual pathway and the distinct functional characteristics of each paradigm.

## Acknowledgments

I am most grateful to the participants. I am also very grateful to the anonymous reviewers for their valuable and insightful comments and suggestions.

## Author Contributions

**Conceptualization:** Yasuhiro Takeshima.

**Data curation:** Yasuhiro Takeshima.

**Formal analysis:** Yasuhiro Takeshima.

**Funding acquisition:** Yasuhiro Takeshima.

**Investigation:** Yasuhiro Takeshima.

**Methodology:** Yasuhiro Takeshima.

**Project administration:** Yasuhiro Takeshima.

**Resources:** Yasuhiro Takeshima.

**Software:** Yasuhiro Takeshima.

**Supervision:** Yasuhiro Takeshima.

**Validation:** Yasuhiro Takeshima.

**Visualization:** Yasuhiro Takeshima.

**Writing – original draft:** Yasuhiro Takeshima.

**Writing – review & editing:** Yasuhiro Takeshima.

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
