## [Decision Letter · Decision Letter 0]

24 Mar 2021

PONE-D-20-40727

The effects of temporal characteristics depend on visual fields for the temporal synchrony processing of audio-visual stimuli

PLOS ONE

Dear Dr. Takeshima,

Thank you for submitting your manuscript to PLOS ONE. After careful consideration, we feel that it has merit but does not fully meet PLOS ONE’s publication criteria as it currently stands. Therefore, we invite you to submit a revised version of the manuscript that addresses the points raised during the review process.

We look forward to receiving your revised manuscript.

Kind regards,

Deborah Apthorp, Ph.D

Academic Editor

PLOS ONE

Additional Editor Comments:

The reviewers have provided careful and detailed reviews. Please pay particular attention to the PLoS One data availability policy and be sure to have all your data (and, preferably, your analysis code) available before submitting your revision. In addition, both reviewers have pointed out that a more careful explanation of the theoretical motivation for the study is required.

Journal Requirements:

2. In line with PLOS' guidelines on detailed reporting (https://journals.plos.org/plosone/s/criteria-for-publication#loc-3), please ensure that you have provided sufficient detail on participant recruitment in the Methods section, including from where participants were recruited.

Reviewers' comments:

Reviewer's Responses to Questions

**Comments to the Author**

1. Is the manuscript technically sound, and do the data support the conclusions?

Reviewer #1: Yes

Reviewer #2: Partly

2. Has the statistical analysis been performed appropriately and rigorously? 

Reviewer #1: Yes

Reviewer #2: No

3. Have the authors made all data underlying the findings in their manuscript fully available?

Reviewer #1: Yes

Reviewer #2: No

4. Is the manuscript presented in an intelligible fashion and written in standard English?

Reviewer #1: Yes

Reviewer #2: Yes

5. Review Comments to the Author

Reviewer #1: The manuscript describes the results of three psychophysical experiments that investigate the effects of visual field eccentricity (central vs peripheral) on audio-visual timing performance.

Each experiment uses a different AV timing task to derive an estimate of the point of subjective simultaneity and temporal bandwidth. Each task also uses serial dependency analysis to determine evidence of temporal recalibration of the kind observed previously (e.g. Van der Burg, et al.).

Experiment 1 uses a simultaneity judgement, Experiment 2 uses temporal order judgement and Experiment 3 uses a stream-bounce judgement, all of which have been used extensively to derive PSS and temporal bandwidths.

Each experiment shows a diverse set of visual field effects, notably that:

bandwidths are broader in central than in peripheral visual field locations for SJs but not TOJs nor SBJ;. Regarding serial dependencies, a visual field effect is observed only for TOJs, and SBJs, not SJs.

To account for this diverse set of results, the authors appeal to differences in visual timing precision previously reported to occur across the visual field. Unfortunately, this account is far too vague to offer any explanatory value, nor is it able to account for the full range of results.

Whilst the experiments appear to be well-conducted, the analyses appropriate, and the results are interesting, practically no theoretical motivation is provided for these experiments. Moreover, few if any hypotheses are offered. Consequently, what the reader is left with is a set of parametric experiments with a diverse set of results, with no meaningful interpretation.

Specific comments

Page 2, Lines 11-13

"...exhibited more auditory preceding timing in the peripheral....."

Not clear what this means. Please rephrase.

"On the other hand...." Other hand relative to what? Consider deleting this cliche

“temporal binding window was larger…”

Presumably, this refers to a broader subjective simultaneity bandwidth, not amplitude. Please specify.

Line 21-23

I don't follow this.

Which visual features are the authors referring to? Temporal response? If so, what kind of temporal response exactly?

Page 4 Line 18-23

The authors need to explain for the reader why they are comparing simultaneity judgment with TOJ tasks. The paragraph (lines 18-23) doesn't provide a justification.

Line 22

‘view distances…”

Presumably, the authors are referring to variations in eccentricity arising from variations in viewing distance of an otherwise identical stimulus. This needs to be clarified.

Page 5 Line 7

It's not clear from the Introduction what the theoretical motivation for the visual eccentricity (visual field) manipulation is.

There are no obvious hypotheses pertaining to this manipulation for any of the three tasks or for the effects of temporal recalibration

Page 10, Lines 19-21

Reference to neural transmission speed is an interpretation, and possibly a prediction/hypothesis. Either way, this doesn't belong in the Results section.

“…a vision leads response is suppressed…’ [italics added]

Again, this is an interpretation, not a description of the experimental Results

Page 14, Lines 6-7

Again, this is an interpretation, not a description of the experimental Results

General Discussion

Page 20, Lines 12-13

How do you know they are more suppressed? It is more descriptive to say that TOJ vision leads responses are less frequent in the visual periphery than in central vision. To invoke suppression implies a mechanism for which these experiments offer no evidence

Lines 14-15

Presumably, the authors are referring to the observation that SJ derived TBWs are broader at central than at peripheral visual field locations.

Page 22, Lines 22-23

I don't follow this sentence. Which temporal characteristics exactly are the authors referring to?

Nor is it clear how the temporal characteristics of the visual field ought to explain the diverse pattern of task dependent visual field effects reported here.

An explicit assumption repeated several times throughout the manuscript is that the peripheral visual field affords higher temporal acuity than the central visual field. Whilst there is neurophsyiological evidence for this, psychophysical evidence is more equivocal. In fact, it's only been convincingly demonstrated using using the critical flicker fusion paradigm, and not other methods - see

https://www.ncbi.nlm.nih.gov/pmc/articles/PMC6264386/

for a nice summary.

Reviewer #2: The current study examined audiovisual simultaneity perception in the central and peripheral visual fields (VFs) using three well-established paradigms: temporal order judgments (TOJ), simultaneity judgments (SJ), and stream/bounce (SB) perception. The PSS was more negative (i.e., in the auditory-leading side) in the center than in the periphery when using TOJ, but no difference when using SJ and SB. The temporal binding window (TBW) was wider in the center than in the periphery when using SJ, but no such difference when using TOJ and SB. Rapid recalibration was observed in terms of PSS in the center when using TOJ, SJ, and SB, and in the periphery when using SJ. In addition, in TOJ, the TBW was narrower when the previous trial had a positive SOA than when it had a negative SOA. These results demonstrated that audiovisual simultaneity perception was partly modulated by the eccentricity of the visual stimuli, but discrepancy occurred when measured using different experimental paradigms.

The current study certainly provides critical measures of human audiovisual simultaneity perception in the center and 10-degree periphery using three methods. That said, I have concerns about the results—whether the data are reliable and whether the explanations are convincing, especially for those task-dependent effects. I also consider the current version of manuscript is too simplified. I therefore have following suggestions and I hope that they can help the author to improve the next version of manuscript:

Major comments:

1. In order to better predict and explain the results observed in the TOJ, SJ, and SB paradigms, the critical differences of their underlying mechanisms should be introduced and discussed. This is especially critical when different results were observed in each paradigm: should these differences be attributed to the different sensory, perceptual, or decisional processing?

As the author mentioned in the Discussion, SJ and TOJ may share the same perceptual mechanism but different decisional processes (e.g., García-Pérez & Alcalá-Quintana, 2015). On the other hand, the SB perception may involve causal relations and attention in addition to audiovisual integration (Shimojo et al., 2001; van Eijk et al., 2008). Hence, stronger theoretical backgrounds will be necessary to understand the distinct results observed in three experiments, and will provide a clearer rationale for the current study.

2. The author proposed an interesting assumption that visual stimulus presented to sustained channel and transient channel may lead to different audiovisual simultaneity perception, because the transient channel has higher temporal resolution and faster processing speed. However, these two characteristics in the temporal processing may be associated with different aspects of audiovisual simultaneity perception. In my intuitive guessing, higher temporal resolution may be associated with narrower audiovisual TBW, whereas the faster processing time may be associated with the shift of PSS (see the model of García-Pérez & Alcalá-Quintana, 2012, and a recent study by Chien et al., 2020). I also found that García-Pérez and Alcalá-Quintana published a paper last year (2020) on this issue.

Hence, I would expect that the PSS should be at more negative (i.e., the auditory stimulus should be presented earlier) in the periphery than in the center, if visual processing is faster in the periphery than in the center. This prediction was not consistent with the results, and even an opposite direction was observed in Experiment 1. I cannot follow the author’s “suppression” account on p. 10.

On the other hand, the result of TBW in Experiment 2 was consistent with my prediction that the TBW was narrower in the periphery than in the center because of the higher temporal resolution in the periphery. However, this result contrasts with trend as a function of eccentricity reported by Stevenson et al. (2012). Please discuss this critical inconsistency.

3. The results of rapid recalibration in SJ and TOJ were in the same direction in the current study, which is inconsistent with Roseboom (2019). On p. 21, if I understand correctly, the author claimed that the effect of rapid recalibration may be cancelled out by the choice-repetition bias in TOJ. However, I do not understand how this explanation is only applied in the periphery rather than in the center, given that the choice-repetition bias should occur in both conditions when they are mixed in a block. In addition, given the fact that response type in TOJ, SJ, and SB are very different from each other, I am also wondering how the choice-repetition bias can be applied in SB but not in SJ.

4. To my knowledge, it is unusual that the rapid recalibration effect is observed in TBW as demonstrated in Experiment 1. Because the rapid recalibration effect in PSS was only observed in the center while the rapid recalibration effect in TBW was observed both in the center and periphery, the explanation on p.11 therefore does not work.

5. I am wondering how the number of participants was determined in each experiment. I can see that the expected number was 20, but only 17 remaining in Experiment 3. I worry that this number is too small (the main effect of modality order was only marginal significant in the rapid recalibration analysis). The SB perception is a subjective experience and therefore a larger individual difference may exist.

6. I have some critical questions regarding the experimental designs:

(1) In Experiment 1, the blank display before the visual target was 500-1000 ms, and the other blank after the visual target was 200-710 ms (this information is missing in the main text). It seems that both blanks were occasionally shorter than the largest SOAs between the visual and auditory target (+/- 510 ms).

(2) Technically, there is no +/-0 ms, only 0 ms. Hence, there should be only 9 SOAs in Experiments 2 and 3. Was the number of trials in the 0 ms SOA doubled as compared to other SOAs?

(3) In Experiment 3, how long is the duration of the motion display? A figure to demonstrate the procedure will be great.

7. More details and clarifications in data analyses are required:

(1) In Experiment 1, please specify how to estimate the PSS and sigma (TBW) based on the cumulative Gaussian function.

(2) In Experiments 2 & 3, please explain the meaning of alpha, PSS and sigma in terms of the Gaussian function, respectively. Were the different alpha scores in the center and periphery critical in Experiment 3?

(3) Did different fitting methods used in Experiment 1 (maximum-likelihood) vs. Experiments 2 & 3 (minimal RMSE) potentially influence the estimation of PSS and sigma?

(4) On p. 19, line 4, were the p value of the two t tests corrected? Were they one- or two-tailed?

8. Based on the results, which paradigm will be recommended for researchers to explore similar issue in the future studies?

Minor comments:

1. The current title is wordy—it should be condensed.

2. The sound was presented from headphone, so the perceived location was near the participant’s head rather than in the front. Does this influence the results of PSS, TWB, and rapid recalibration since the visual and auditory stimuli were spatially separate (i.e., violating the unity assumption)?

3. Figures 4 & 6: It would be easier to read if the same conditions have the same color code (such as in Figure 2, the center condition has open dots in (a) and white bars in (b,c)). Same suggestion for Figures 3, 5, & 7.

4. A table summarizing all results of three experiments will be helpful.

References

Chien, S. E., Chen, Y. C., Matsumoto, A., Yamashita, W., Shih, K. T., Tsujimura, S. I., & Yeh, S. L. (2020). The modulation of background color on perceiving audiovisual simultaneity. Vision research, 172, 1-10.

García-Pérez, M. A., & Alcalá-Quintana, R. (2012). On the discrepant results in synchrony judgment and temporal-order judgment tasks: a quantitative model. Psychonomic bulletin & review, 19(5), 820-846.

García-Pérez, M. A., & Alcala-Quintana, R. (2020). Assessing multisensory integration and estimating speed of processing with the dual-presentation timing task: Model and data. Journal of Mathematical Psychology, 96, 102351.

Shimojo, S., Scheier, C., Nijhawan, R., Shams, L., Kamitani, Y., & Watanabe, K. (2001). Beyond perceptual modality: Auditory effects on visual perception. Acoustical Science and Technology, 22(2), 61-67.

Stevenson, R. A., Fister, J. K., Barnett, Z. P., Nidiffer, A. R., & Wallace, M. T. (2012). Interactions between the spatial and temporal stimulus factors that influence multisensory integration in human performance. Experimental Brain Research, 219(1), 121-137.

6. PLOS authors have the option to publish the peer review history of their article (what does this mean?). If published, this will include your full peer review and any attached files.

Reviewer #1: No

Reviewer #2: No

---

## [Author Response · Author response to Decision Letter 0]

6 May 2021

Replies to Reviewer 1’s comments

I appreciate your helpful and valuable comments on this manuscript. I have revised the manuscript to clarify the theoretical motivation and rationale interpretation of this study.

Point 1: Page 2, Lines 11-13

"...exhibited more auditory preceding timing in the peripheral....."

Not clear what this means. Please rephrase.

Reply: I have rephrased this description to clarify the meaning of the text (page 2, lines 22-25).

“The results indicate that auditory stimuli should be presented earlier for visual stimuli in the central visual field than in the peripheral visual field condition for one to perceive subjective simultaneity in the temporal order judgment task conducted in this study.”

Point 2: "On the other hand...." Other hand relative to what? Consider deleting this cliche

Reply: I have deleted the pertinent text.

Point 3: “temporal binding window was larger…”

Presumably, this refers to a broader subjective simultaneity bandwidth, not amplitude. Please specify.

Reply: I revised this statement as per your suggestion (page 2, lines 25-27).

“Meanwhile, the subjective simultaneity bandwidth was broader in the central visual field than in the peripheral visual field during the simultaneity judgment task.”

Point 4: Line 21-23

I don't follow this. Which visual features are the authors referring to? Temporal response? If so, what kind of temporal response exactly?

Reply: The purpose of this study was to investigate the effects of differences in visual processing speed (including temporal resolution) based on visual fields on temporal synchrony perception of audio-visual stimuli. I have revised the passage to clarify the purpose (page 2, lines 31-35).

“These results suggest that differences in visual processing speed based on the visual field modulate the temporal processing of audio-visual stimuli. Future studies are necessary to confirm the effects of compensation regarding differences in the temporal resolution of the visual filed in later cortical visual pathway on visual field differences in audio-visual temporal synchrony.”

Point 5: Page 4 Line 18-23

The authors need to explain for the reader why they are comparing simultaneity judgment with TOJ tasks. The paragraph (lines 18-23) doesn't provide a justification.

Reply: I have added explanations as to why SJ was compared with TOJ tasks (page 5, lines 86-89).

“Van Eijk, Kohlrausch, Juola, and van de Par [26] have shown a lack of correlation between SJ and TOJ PSS, and proposed a different kind of sensitivity between TOJ and SJ for audio-visual asynchrony. Therefore, it is necessary to use both TOJ and SJ tasks to examine audio-visual temporal synchrony.”

Point 6: Line 22

‘view distances…”

Presumably, the authors are referring to variations in eccentricity arising from variations in viewing distance of an otherwise identical stimulus. This needs to be clarified.

Reply: In Kopinska and Harris’s study, eccentricity and viewing distance were manipulated individually. I have revised the descriptions to clarify this point (page 4, line 80-page 5, line 83).

“Kopinska and Harris [11] compared the PSS between central (0° eccentricity) and peripheral (20° eccentricity) VFs using the TOJ task by manipulating both the eccentricity of the visual stimulus and participants’ viewing distance and showed that the timing of subjective simultaneity did not differ between VFs.”

Point 7: Page 5 Line 7

It's not clear from the Introduction what the theoretical motivation for the visual eccentricity (visual field) manipulation is.

Reply: My previous study reported that the difference in visual processing speed based on spatial frequency modulates audio-visual temporal synchrony perception by using the SJ task. Therefore, this study was conducted to confirm this finding by manipulating the eccentricity of visual stimuli, which is also associated with visual processing speed. I have added the descriptions about this theoretical motivation (page 4, lines 64-78).

“Temporal synchrony perception for audio-visual stimuli is affected by the processing speed of the visual system. Previous studies have reported that the PSS for audio-visual stimuli was directed to more auditory leads in low spatial frequency stimuli than in high spatial frequency stimuli [20, 21]. Visual systems have two channels, namely the transient and sustained channels, with each kind having different temporal resolutions (processing speeds) [22]. Transient channels respond to low spatial frequencies and exhibit high temporal resolution, whereas sustained channels exhibit high spatial frequencies and low temporal resolution [23]. Therefore, auditory stimuli should be presented earlier for low spatial frequency stimuli than for high spatial frequency stimuli for one to perceive subjective simultaneity. These two channels are projected from different retinal positions [24]. The transient channel is projected from the peripheral visual field (VF), while the sustained channel is projected from the central VF; thus, the processing speed of visual stimuli in the central VF is slower than that in the peripheral VF. The present study confirmed the effects of visual temporal resolution on audio-visual synchrony perception by manipulating the VF in which the visual stimulus is presented (i.e., eccentricity). “

Point 8: There are no obvious hypotheses pertaining to this manipulation for any of the three tasks or for the effects of temporal recalibration

Reply: I did not set obvious hypotheses for any of the three tasks since the purpose of this study was to confirm the differences in PSS due to the visual processing speed. For rapid recalibration, I added some predictions in the revised manuscript. (page 5, line 102-page 6, line 108).

“Using the SJ task, Takeshima [30] reported that normal rapid temporal recalibration occurred regardless of a difference in visual processing speed based on spatial frequency. The present study predicts that normal rapid temporal recalibration would be observed in both the central and peripheral VFs in the SJ task. In the TOJ task, as in Roseboom [29], temporal recalibration in the direction opposite to that in the SJ task is predicted to occur. This study also explores rapid temporal recalibration using SB perception since previous studies have not examined this using the paradigm under investigation. “

Point 9: Page 10, Lines 19-21

Reference to neural transmission speed is an interpretation, and possibly a prediction/hypothesis. Either way, this doesn't belong in the Results section.

“…a vision leads response is suppressed…’ [italics added]

Again, this is an interpretation, not a description of the experimental Results

Page 14, Lines 6-7

Again, this is an interpretation, not a description of the experimental Results

Reply: In the revised manuscript, I have clearly divided the “Results and discussion” section of the previous manuscript into a “Results” section and a “Discussion” section in each experiment.

Point 10: Page 20, Lines 12-13

How do you know they are more suppressed? It is more descriptive to say that TOJ vision leads responses are less frequent in the visual periphery than in central vision. To invoke suppression implies a mechanism for which these experiments offer no evidence

Reply: Although the result of the TOJ task was inconsistent with the VF difference in temporal resolution, it was consistent with that in visual latency in previous studies. I have discussed this point in the revised manuscript (page 12, line 245-255).

“The difference observed between the central and peripheral VFs was opposite to that predicted by the difference in temporal resolution. The temporal resolution is higher for the peripheral VF than for the central VF [24]; thus, it was predicted that auditory stimuli should be presented earlier for visual stimuli in the peripheral VF than in the central VF condition for one to perceive subjective simultaneity. However, the current observed difference in the PSS was consistent with the difference in visual latency between the central and peripheral VFs. The response time for the visual stimulus was shorter at the central VF than at the peripheral VF [35]. A previous study that manipulae spatial frequency also observed differences in the PSS that were consistent with differences observed in response time [20, 30]. Therefore, in a TOJ task, differences in the PSS between central and peripheral VFs could be attributed to differences in the response time, not temporal resolution.”

Point 11: Page 20, Lines 12-13

How do you know they are more suppressed? It is more descriptive to say that TOJ vision leads responses are less frequent in the visual periphery than in central vision. To invoke suppression implies a mechanism for which these experiments offer no evidence

Reply: I have revised the implications for the description of the PSS value (page 23, line 512-513).

“Experiment 1 showed that the PSS value was smaller in the central VF condition than in the peripheral VF condition.”

Point 12: Lines 14-15

Presumably, the authors are referring to the observation that SJ derived TBWs are broader at central than at peripheral visual field locations.

Reply: I have revised the sentence to refer to the VF difference in TBW width (page 23, lines 514-515).

“Experiment 2 indicated that the TBW was broader at the central VF than at the peripheral VF.”

Point 13: Page 22, Lines 22-23

I don't follow this sentence. Which temporal characteristics exactly are the authors referring to?

Reply: I have revised the last paragraph of the General discussion section to clarify the claim of this study (page 26, lines 561-567).

“The present findings also suggest a discrepancy in judgment between the VFs; judging simultaneity between visual events and sound in the central VF, whereas judging bounce induced by sound in the peripheral VFs. The use of SB perception in investigating temporal synchrony perception for audio-visual stimuli needs to be carefully considered. Moreover, bounce responses differ between the VFs (i.e., the results of a no-sound experiment and alpha values). It is necessary to further investigate whether this difference affects the estimation of the PSS and TBW of SB perception.”

Point 14: Nor is it clear how the temporal characteristics of the visual field ought to explain the diverse pattern of task dependent visual field effects reported here.

Reply: Thank you for your comment. Based on this, I have reinterpreted the results regarding the VF difference in this study (page 24, line 523-page 25, line 539).

“VF differences in audio-visual temporal synchrony were observed in the TOJ and SJ tasks. In a TOJ task, the auditory stimulus is presented earlier as a visual stimulus for one to perceive subjective simultaneity in the central than in the peripheral VF condition. In an SJ task, the TBW width of the central VF is broader than that of the peripheral VF. The difference in the PSS of the TOJ task was consistent with the VF difference in a visual latency, whereas the difference in the TBW of the SJ task was consistent with the VF difference in a temporal resolution. As a preliminary prediction, the VF differences in the PSS and TBW both follow differences in temporal resolution. It has been shown that the difference in eccentricity-dependent temporal resolution are observed in the early visual cortex and are compensated later in the cortical visual pathway [39]. A TOJ task is assumed to be associated with higher-order processing compared with an SJ task [26, 40]. Therefore, the discrepancy between VF difference in the PSS and temporal resolution could be attributed to the visual hierarchy of eccentricity-dependent temporal contrast in a TOJ task. Moreover, a TOJ task has been proposed to be sensitive to PSS change [26], whereas an SJ task has been proposed to be sensitive to TBW change [41]. Thus, the discrepancy in the indices for which VF differences were observed between the TOJ and SJ tasks would reflect the difference in sensitivity of indices to audio-visual synchrony perception.” 

Replies to Reviewer 2’s comments

I appreciate your helpful and valuable comments on this manuscript and have revised it based on your comments.

Major comments:

1. In order to better predict and explain the results observed in the TOJ, SJ, and SB paradigms, the critical differences of their underlying mechanisms should be introduced and discussed. This is especially critical when different results were observed in each paradigm: should these differences be attributed to the different sensory, perceptual, or decisional processing?

As the author mentioned in the Discussion, SJ and TOJ may share the same perceptual mechanism but different decisional processes (e.g., García-Pérez & Alcalá-Quintana, 2015). On the other hand, the SB perception may involve causal relations and attention in addition to audiovisual integration (Shimojo et al., 2001; van Eijk et al., 2008). Hence, stronger theoretical backgrounds will be necessary to understand the distinct results observed in three experiments, and will provide a clearer rationale for the current study.

Reply: I have reinterpreted the results of the VF difference between the TOJ and SJ tasks based on the compensation of the difference in temporal resolution of the visual filed in later cortical visual pathway (page 24, line 525-page 25, line 539).

“VF differences in audio-visual temporal synchrony were observed in the TOJ and SJ tasks. In a TOJ task, the auditory stimulus is presented earlier as a visual stimulus for one to perceive subjective simultaneity in the central than in the peripheral VF condition. In an SJ task, the TBW width of the central VF is broader than that of the peripheral VF. The difference in the PSS of the TOJ task was consistent with the VF difference in a visual latency, whereas the difference in the TBW of the SJ task was consistent with the VF difference in a temporal resolution. As a preliminary prediction, the VF differences in the PSS and TBW both follow differences in temporal resolution. It has been shown that the difference in eccentricity-dependent temporal resolution are observed in the early visual cortex and are compensated later in the cortical visual pathway [39]. A TOJ task is assumed to be associated with higher-order processing compared with an SJ task [26, 40]. Therefore, the discrepancy between VF difference in the PSS and temporal resolution could be attributed to the visual hierarchy of eccentricity-dependent temporal contrast in a TOJ task. Moreover, a TOJ task has been proposed to be sensitive to PSS change [26], whereas an SJ task has been proposed to be sensitive to TBW change [41]. Thus, the discrepancy in the indices for which VF differences were observed between the TOJ and SJ tasks would reflect the difference in sensitivity of indices to audio-visual synchrony perception.”

2. The author proposed an interesting assumption that visual stimulus presented to sustained channel and transient channel may lead to different audiovisual simultaneity perception, because the transient channel has higher temporal resolution and faster processing speed. However, these two characteristics in the temporal processing may be associated with different aspects of audiovisual simultaneity perception. In my intuitive guessing, higher temporal resolution may be associated with narrower audiovisual TBW, whereas the faster processing time may be associated with the shift of PSS (see the model of García-Pérez & Alcalá-Quintana, 2012, and a recent study by Chien et al., 2020). I also found that García-Pérez and Alcalá-Quintana published a paper last year (2020) on this issue.

Hence, I would expect that the PSS should be at more negative (i.e., the auditory stimulus should be presented earlier) in the periphery than in the center, if visual processing is faster in the periphery than in the center. This prediction was not consistent with the results, and even an opposite direction was observed in Experiment 1. I cannot follow the author’s “suppression” account on p. 10.

On the other hand, the result of TBW in Experiment 2 was consistent with my prediction that the TBW was narrower in the periphery than in the center because of the higher temporal resolution in the periphery. However, this result contrasts with trend as a function of eccentricity reported by Stevenson et al. (2012). Please discuss this critical inconsistency.

Reply: Visual latency is shorter in the central VF than in the peripheral VF, whereas temporal resolution is higher in the peripheral VF than in the central VF. The PSS result in the TOJ task was consistent with the prediction because of this visual latency difference. Thus, I have reinterpreted the PSS results in Experiment 1 (page 12, lines 245-255).

“The difference observed between the central and peripheral VFs was opposite to that predicted by the difference in temporal resolution. The temporal resolution is higher for the peripheral VF than for the central VF [24]; thus, it was predicted that auditory stimuli should be presented earlier for visual stimuli in the peripheral VF than in the central VF condition for one to perceive subjective simultaneity. However, the current observed difference in the PSS was consistent with the difference in visual latency between the central and peripheral VFs. The response time for the visual stimulus was shorter at the central VF than at the peripheral VF [35]. A previous study that manipulae spatial frequency also observed differences in the PSS that were consistent with differences observed in response time [20, 30]. Therefore, in a TOJ task, differences in the PSS between central and peripheral VFs could be attributed to differences in the response time, not temporal resolution.”

Stevenson et al. (2012) reported an opposite TBW difference from the present study. However, their study manipulated a wider range of eccentricities, and a similar PSS difference to the results of Experiment 2 was observed at eccentricities closer to the present study. Therefore, their results are considered to be consistent with the results of this study (page 17, line 376-page 18, line 383).

“Stevenson, Kruger Fister, Barnett, Nidiffer, and Wallace [37] have found a broader TBW in the peripheral VF than in the central VF. However, a higher percentage of simultaneous judgment was observed in the peripheral VF outside of the present experiment (60° and 90° eccentricities) in their study. In the peripheral VF (30° eccentricity), which was closer to the present experiment, the percentage of simultaneous judgment is slightly lower than that in the central VF. Therefore, the TBW of the central VF is assumed to become wider than that of the peripheral VF up to a certain range (at least 30° eccentricity), which is attributed to temporal resolution.”

3. The results of rapid recalibration in SJ and TOJ were in the same direction in the current study, which is inconsistent with Roseboom (2019). On p. 21, if I understand correctly, the author claimed that the effect of rapid recalibration may be cancelled out by the choice-repetition bias in TOJ. However, I do not understand how this explanation is only applied in the periphery rather than in the center, given that the choice-repetition bias should occur in both conditions when they are mixed in a block. In addition, given the fact that response type in TOJ, SJ, and SB are very different from each other, I am also wondering how the choice-repetition bias can be applied in SB but not in SJ.

Reply: Previous studies reporting choice-repetition bias in a TOJ task used blurred visual stimuli. Although the present study used a shape-edge visual stimulus, this stimulus was presumably perceived as blurred in the peripheral VF. Thus, a consistent interpretation can be made if the choice-repetition bias occurs only for blurred visual stimuli (page 12, line 256-page 13, line 269).

“The results of normal rapid recalibration for the central VF were inconsistent with the results of previous studies. Roseboom [29] showed that the PSSs shifted in opposite directions from normal rapid temporal recalibration in a TOJ task. Moreover, Keane, Bland, Matthews, Carroll, and Wallis [36] found that opposite-directed PSS shifts were induced by choice-repetition bias in a TOJ task. Choice-repetition bias refers to the tendency to repeat judgments of temporal order of a previous trial on a current trial. Additionally, rapid recalibration was obfuscated by opposite-directed PSS shifts due to a choice-repetition bias [36]. In this study, the choice-repetition bias did not occur only in the central VF. Roseboom [29] and Keane et al. [36] also presented the visual stimulus in the central VF; however, Roseboom [29] used a Gaussian probe and Keane et al. [36] used a small light-emitting diode. Unlike the circle used in this experiment, these visual stimuli had blurred edges. In the present experiment, choice-repetition bias also occurred in the peripheral VF, in which the visual stimuli were perceived as blurred. Therefore, blurred visual stimuli would be necessary for occurring choice-repetition bias in the TOJ task for audio-visual stimuli.”

In Experiment 3, I assumed that the difference in the judged information between the central and peripheral VFs caused the difference in rapid recalibration. Vroomen and Keetels (2020) showed that the PSS of SJ becomes positive, whereas that of SB becomes negative. Although a significant difference was not observed, the PSS of central VF was positive and that of the peripheral VF was negative. Thus, a consistent interpretation can be made if SB perception does not induce rapid recalibration (page 22, line 490-page 23, line 506).

“The PSS values did not significantly differ between the central and peripheral VFs in SB perception. Although there was no significant difference, the signs of PSS values were different between the central and peripheral VFs, a positive value in the central VF and a negative value in the peripheral VF. Vroomen and Keetels [38] showed a discrepancy between the optimal time of sound to induce bounce and maximal audio-visual synchrony in SB displays. The former timing was before the instance of superposition (i.e., negative value), whereas the later timing was after the instance of superposition (i.e., positive value). Thus, in this experiment, participants may have judged the simultaneity between visual events (i.e., contact between two moving visual stimuli) and sound in the central VF, whereas they judged the bounce perception of the two visual stimuli induced by sound in the peripheral VF.

 The discrepancy in results between the central and peripheral VF in rapid recalibration could be explained by this difference in judgment. In this experiment, normal rapid recalibration occurred only in the central VF and not in the peripheral VF. The occurrence of rapid recalibration due to SB perception has not been examined. If SB judgment does not cause a change in the PSS due to the timing of sound on a previous trial, it is consistent with results of this experiment. Future studies are needed to confirm this postulation, since there are little experimental data on rapid recalibration using SB perception.”

4. To my knowledge, it is unusual that the rapid recalibration effect is observed in TBW as demonstrated in Experiment 1. Because the rapid recalibration effect in PSS was only observed in the center while the rapid recalibration effect in TBW was observed both in the center and periphery, the explanation on p.11 therefore does not work.

Reply: As you have pointed out, the TBW shift during rapid recalibration is unusual. I have also not predicted and interpreted this result. First, few data on TBW have been reported in previous studies of rapid recalibration. Therefore, I would like to elucidate the mechanism underlying this effect in the future. I have added description about this point (page 13, lines 270-278).

“Moreover, the difference in the TBW between audition and vision leads during the rapid recalibration process is a novel finding. In Experiment 1, the TBW width narrowed in the preceding vision leads presentation than in the preceding audition leads presentation. A narrow TBW indicates a high sensitivity to judge the temporal order between visual and auditory stimuli. Therefore, this finding shows that the temporal information of visual precedence in a previous trial increase the sensitivity in a TOJ task for audio-visual stimuli. In previous studies of rapid recalibration (e.g., [18], [29], and [34]), the difference in TBW width between audition and vision leads conditions have not been investigated. Such a difference in the TBW was not predicted in this study, and this needs to be examined in more detail in the future.”

5. I am wondering how the number of participants was determined in each experiment. I can see that the expected number was 20, but only 17 remaining in Experiment 3. I worry that this number is too small (the main effect of modality order was only marginal significant in the rapid recalibration analysis). The SB perception is a subjective experience and therefore a larger individual difference may exist.

Reply: The sample size was determined using PANGEA. The result of the calculation showed that the power (1-β) was 0.86 with a sample size of 16 participants in each experiment. I also felt that there were many outliers and the sample size became smaller in Experiment 3; however, I did not consider this to have such a large impact.

6. I have some critical questions regarding the experimental designs:

(1) In Experiment 1, the blank display before the visual target was 500-1000 ms, and the other blank after the visual target was 200-710 ms (this information is missing in the main text). It seems that both blanks were occasionally shorter than the largest SOAs between the visual and auditory target (+/- 510 ms).

(2) Technically, there is no +/-0 ms, only 0 ms. Hence, there should be only 9 SOAs in Experiments 2 and 3. Was the number of trials in the 0 ms SOA doubled as compared to other SOAs?

(3) In Experiment 3, how long is the duration of the motion display? A figure to demonstrate the procedure will be great.

Reply: (1) I have added a detailed description of the duration of the blank display (page 8, lines 160-167).

“The duration of this blank display was a fixed length of 500 ms, plus an additional SOA when the auditory stimulus was presented prior to the visual stimulus (i.e., 500 ms at the shortest and 1010 ms at the longest). During the target display period, a circle was presented at one of the following three locations: center, left, or right. The tone either preceded or followed the onset of the circle using the SOA that had been drawn randomly from the set. After the target display, a blank display was again presented. The duration of this blank display was a fixed length of 200 ms, plus an additional SOA when the visual stimulus was presented prior to the auditory stimulus (i.e., 200 ms at the shortest and 710 ms at the longest).”

Reply: (2) As you have pointed out, the number of trials in the 0 ms SOA doubled compared with other SOAs. I have added descriptions about this point (page 14, lines 305-307).

“One block comprised 80 trials, with 10 trials for each SOA condition (i.e., 16 trials for 0 ms SOA and 8 trials for other SOAs).”

Reply: (3) The duration of motion display was 1440 ms in the experiment. I have added descriptions and figure about this point (page 19, lines 417-418).

“Each trial consisted of a 500 ms fixation stimulus followed by motion displays (a duration of 1440 ms, see Fig 6).”

7. More details and clarifications in data analyses are required:

(1) In Experiment 1, please specify how to estimate the PSS and sigma (TBW) based on the cumulative Gaussian function.

(2) In Experiments 2 & 3, please explain the meaning of alpha, PSS and sigma in terms of the Gaussian function, respectively. Were the different alpha scores in the center and periphery critical in Experiment 3?

(3) Did different fitting methods used in Experiment 1 (maximum-likelihood) vs. Experiments 2 & 3 (minimal RMSE) potentially influence the estimation of PSS and sigma?

(4) On p. 19, line 4, were the p value of the two t tests corrected? Were they one- or two-tailed?

Reply: (1) I have added the formula for the cumulative Gaussian function in the revised manuscript (page 9, line 184).

P (response|SOA) = 1/(1+e^[-1/sigma (SOA-PSS)] )

(2) I have added a description of these parameters in terms of the Gaussian function in Experiment 2 (page 15, lines 318-320).

“…these parameters indicated the height, peak position on the SOA axis, and the width of the Gaussian function, respectively.”

The difference in the alpha value indicates the difference in the degree of bounce perception in Experiment 3. However, it is not difficult to determine whether this difference is critical in connection with the results of Experiment 3. I have added a description about this point in the General discussion section (page 26, lines 565-567).

“Moreover, bounce responses differ between the VFs (i.e., the results of a no-sound experiment and alpha values). It is necessary to further investigate whether this difference affects the estimation of the PSS and TBW of SB perception.”

(3) I also calculated the RMSE of fitting in Experiment 1 and these values did not largely differ from the values of Experiments 2 and 3. Therefore, I assume that such a difference in the fitting method would not influence the estimations.

(4) I used the two-tailed one-sample t-tests, and these tests and p values were corrected. I apologize for inserting the incorrect figure regarding these results (Fig 8) in the previous manuscript. I have revised the figure.

8. Based on the results, which paradigm will be recommended for researchers to explore similar issue in the future studies?

Reply: I recommend using the explicit paradigm (i.e., TOJ or SJ task), since it was suggested that different judgments occur between VFs in the SB perception based on this study. I have added at description of this point in the revised manuscript (page 27, lines 590-595).

“Furthermore, a paradigm difference in temporal synchrony perception was also observed in this study, among TOJ, SJ, and SB. The present findings suggest that explicit measurements (i.e., TOJ and SJ tasks) are more desirable to examine the temporal synchrony perception of audio-visual stimuli between VFs, because of the different judgments that occur when using an implicit measurement (i.e., SB perception).”

Minor comments:

1. The current title is wordy—it should be condensed.

Reply: I have revised the manuscript title per your suggestion.

2. The sound was presented from headphone, so the perceived location was near the participant’s head rather than in the front. Does this influence the results of PSS, TWB, and rapid recalibration since the visual and auditory stimuli were spatially separate (i.e., violating the unity assumption)?

Reply: The distance between the participant and the display was approximately 70 cm. Many previous studies investigating the PSS, the TBW, and rapid recalibration have also used headphones to present sound to participants. Therefore, I assume that the use of headphones does not influence these results.

3. Figures 4 & 6: It would be easier to read if the same conditions have the same color code (such as in Figure 2, the center condition has open dots in (a) and white bars in (b,c)). Same suggestion for Figures 3, 5, & 7.

Reply: I have revised these figures per your suggestion.

4. A table summarizing all results of three experiments will be helpful.

Reply: I have added a table summarizing the results of the experiments per your suggestion.

---

## [Decision Letter · Decision Letter 1]

22 Jun 2021

PONE-D-20-40727R1

Visual field differences in temporal synchrony processing for audio-visual stimuli

PLOS ONE

Dear Dr. Takeshima,

Thank you for submitting your manuscript to PLOS ONE. After careful consideration, we feel that it has merit but does not fully meet PLOS ONE’s publication criteria as it currently stands. Therefore, we invite you to submit a revised version of the manuscript that addresses the points raised during the review process.

We look forward to receiving your revised manuscript.

Kind regards,

Deborah Apthorp, Ph.D

Academic Editor

PLOS ONE

Additional Editor Comments (if provided):

Thank you for your revisions to your manuscript. While the reviewers consider the manuscript considerably improved, several important concerns remain; in particular, both reviewers find that the English in the paper needs improving for comprehensibility, and both ask for clearer and more concise theoretical justifications for the experiments, as well as clearer explanations of the results.

Please revise the paper according to both reviewers' concerns.

Reviewers' comments:

Reviewer's Responses to Questions

**Comments to the Author**

1. If the authors have adequately addressed your comments raised in a previous round of review and you feel that this manuscript is now acceptable for publication, you may indicate that here to bypass the “Comments to the Author” section, enter your conflict of interest statement in the “Confidential to Editor” section, and submit your "Accept" recommendation.

Reviewer #1: (No Response)

Reviewer #2: (No Response)

2. Is the manuscript technically sound, and do the data support the conclusions?

Reviewer #1: Yes

Reviewer #2: Partly

3. Has the statistical analysis been performed appropriately and rigorously? 

Reviewer #1: Yes

Reviewer #2: Yes

4. Have the authors made all data underlying the findings in their manuscript fully available?

Reviewer #1: Yes

Reviewer #2: (No Response)

5. Is the manuscript presented in an intelligible fashion and written in standard English?

Reviewer #1: No

Reviewer #2: No

6. Review Comments to the Author

Reviewer #1: General comments:

The authors have done a good job addressing many of my previous concerns. The paper is substantially improved. However, some of my earlier concerns remain.

In particular, the Introduction. The authors need to explain why they have chosen to compare these various psychophysical measures of temporal synchrony. At present, it reads like a parametric list. That can be fine, but it needs to be theoretically, or at least empirically contextualised.

I also strongly encourage the authors to seek advice on English grammar before resubmitting.

There are numerous instances where tense confusions, awkward and/or imprecise phrasing leaves this reader confused.

As for its scientific contribution, the paper offers some interesting and novel findings, particularly the interaction between eccentricity and modality order in the temporal recalibration experiments. Unfortunately, the absence of a coherent explanation makes it difficult to evaluate the significance of these findings. Perhaps the observation fact that these different measures AV synchrony perception exhibit distinct functional characteristics is the paper’s major contribution.

Beyond this, I don’t know what else to suggest.

Specific comments:

Abstract

The authors mention the three methods they use to measure audio-visual timing performance. No explanation or motivation is provided as to why one might use different measures, or what they actually refer to.

The authors correctly describe the main findings associated with each of the various tasks used in this paper. No attempt is offered in the abstract to interpret these task-contingent effects.

Why use these different tasks? I would like to see some attempt to understand the variety of effects observed across the different tasks

Introduction.

39-42

Why do the authors make reference to this asymmetry? It doesn't seem relevant to the point they're making about the potential functional significance of AV synchrony

59-60

Of greatest relevance here is that a prolonged period of adaptation is not necessary to observe temporal recalibration.

67

...are composed of at least two spatio-temporal channels.

73-75

This is an over-generalisation. Please tone this back

Discussion

240

Process should be Processes

252

Spelling - manipulate

268-269

I don't follow what the authors mean here

Are they saying that blurred edges are a necessary constraint?

Table 1 title

Summary of the three experiments' main findings

524

Do the authors mean "In our TOJ task"?

I don't follow this sentence

529-530

Can the authors please clarify this statement?

532

"A TOJ task"

Suggest: TOJ tasks are assumed to be...

535

"a TOJ task"

Suggest:

TOJ tasks are proposed

562-563

Please clarify. I don't see how this relates to the previous sentence

568

Grammar!

574-575

Suggest: "Therefore, rapid recalibration may be induced by ...."

Reviewer #2: I appreciate that the author has tried to reply to my previous comments and to better explain the current results. However, it seems that each effect in each experimental paradigm is explained by distinct mechanism, and these explanations are sometimes quite speculative and unconvincing. Taken together, even though the current version of manuscript is improved compared to previous version, I would expect that the author provides more parsimonious and coherent explanations. In addition, the readability of the manuscript should be improved, and professional English editing is essential.

1. In my previous point 1, I suggested that the author should introduce the critical differences of the mechanisms underlying the three paradigms used here. This is critical to help the author to predict possible results, and to provide readers clearer scope and motivation for the current study.

2. In my previous point 2, I have proposed that the dissociation between processing time and temporal resolution, which may lead to different predictions for PSS and TBW, respectively. Now the author suggested that the visual latency (which is close to the idea of processing time) is shorter in the center than in the periphery, which can explain the PSS result in the TOJ task. There should be hypothesis based on visual latency as a function of visual field addressed in Introduction.

3. P. 12, Keane et al. (2020) demonstrated the “choice-repetition bias” account based on reanalyzing Roseboom’s (2019) data; Keane et al., instead, reported a null rapid recalibration effect in Experiment 1, and a typical rapid recalibration effect in Experiment 2. Hence, the author’s “blurry visual information” account cannot work.

Following the argument of “choice-repetition bias”, is it possible that the more pronounced rapid recalibration effect in both visual fields was a result of over-estimation due to this bias?

4. Given that the motion display was longer than 1 sec, was the participant’s eye movement monitored or controlled?

5. In Experiment 3, given the fact that the PSS was not significantly different in the center and periphery (p = .69), I don’t see any explanation is needed. Specifically, the author’s suggestion that the participants made different judgments in the center and periphery is not convincing, and so is the following paragraph for the rapid recalibration effect.

6. Table 1: in the summary of rapid recalibration, it is weird to compare the results in the center and periphery either in the A-leading PSS or in the V-leading PSS. The summary should be the comparison between the A-leading PSS and in the V-leading PSS in the center and periphery, respectively.

7. Here are some arguments that I cannot follow:

(1) P. 10, lines 204-205: I do not understand this argument since the results were analyzed and presented separately in the central and peripheral visual field.

(2) P. 25, lines 533-535: If eccentricity effect can be compensated in later visual processing, and the TOJ involved higher-order processing, how could the former explain the PSS difference in the center and periphery in the TOJ?

8. There are still some mistakes in the manuscript:

(1) P. 8, line 170: In Experiment 1, there were 10 SOAs. If there were 80 trials in a block, then there should be only 8 trials for each SOA. Same problem on p. 14, line 306, and p. 19, line 422.

(2) P. 11, line 223, the PSS should be “lower” or “smaller” in the center than in the periphery, since the values were negative.

(3) P. 14, line 298, and p. 19, line 412: it is unclear to me why the author insists to keep the “±0 ms” SOA, since it does not make sense at all.

(4) In the Procedure in Experiment 3, was there also double number of trials at the 0 ms SOA than other SOAs?

(5) P. 28, line 603: “…rapid temporal recalibration occurred only at the periphery VF in the TOJ…” should be at the central VF.

7. PLOS authors have the option to publish the peer review history of their article (what does this mean?). If published, this will include your full peer review and any attached files.

Reviewer #1: No

Reviewer #2: No

---

## [Author Response · Author response to Decision Letter 1]

31 Jul 2021

Replies to Reviewer 1’s comments

I appreciate your helpful and valuable comments on this manuscript. I have revised the manuscript to clarify the theoretical motivation and rationale of this study.

Point 1: Abstract

The authors mention the three methods they use to measure audio-visual timing performance. No explanation or motivation is provided as to why one might use different measures, or what they actually refer to. The authors correctly describe the main findings associated with each of the various tasks used in this paper. No attempt is offered in the abstract to interpret these task-contingent effects. Why use these different tasks? I would like to see some attempt to understand the variety of effects observed across the different tasks.

Reply: I have added the explanation for the use of different measures in this study (lines 22–23), and suggestions from current results (lines 34–36) to the Abstract.

Point 2: 39-42

Why do the authors make reference to this asymmetry? It doesn't seem relevant to the point they're making about the potential functional significance of AV synchrony

Reply: I have deleted the relevant statements based on your comment.

Point 3: 59-60

Of greatest relevance here is that a prolonged period of adaptation is not necessary to observe temporal recalibration.

Reply: I have revised the description as per your suggestion (lines 60–61).

Point 4: 67

...are composed of at least two spatio-temporal channels.

Reply: I have revised the description as per your suggestion (lines 68–69).

Point 5: 73-75

This is an over-generalisation. Please tone this back

Reply: I have revised the relevant statements to be more precise (lines 74–79).

“The retinal positions of vision have a similar difference in these two channels: The central vision has a low temporal resolution, while the peripheral vision has a high temporal resolution [23]. Furthermore, there is a difference in visual latency between the central and peripheral visual fields (VFs): The response time for the visual stimulus is shorter at the central VF than it is at the peripheral VF [24]. Hence, the processing speed differs between the central and peripheral VFs.”

Point 6: 240

Process should be Processes

Reply: I have revised the description as per your suggestion (line 252).

Point 7: 252

Spelling - manipulate

Reply: I apologize for this typo. This has been corrected (line 264).

Point 8: 268-269

I don't follow what the authors mean here

Are they saying that blurred edges are a necessary constraint?

Reply: Reviewer 2 also could not follow the discussion. I have therefore reinterpreted this result on the basis of information reliability (lines 274–278).

“In this study, choice-repetition bias was suppressed in the central VF. Low information reliability induces a larger choice-repetition bias [39, 40]. Therefore, the reliability of judging temporal order for audio-visual stimuli would be high in the central VF. This speculation needs to be further investigated”

Point 9: Table 1 title

Summary of the three experiments' main findings

Reply: I have changed the title of Table 1 as per your suggestion.

Point 10: 524

Do the authors mean "In our TOJ task"? I don't follow this sentence

Reply: I have revised the wording to make it more understandable (line 530-533).

“VF differences in audio-visual temporal synchrony were observed in the TOJ and SJ tasks in this study. In the TOJ task of Experiment 1, the PSS score indicated that the auditory stimulus is presented earlier as a visual stimulus for one to perceive subjective simultaneity in the central than in the peripheral VF condition.”

Point 11: 529-530

Can the authors please clarify this statement?

Reply: I have clarified this statement (lines 536–539).

“As a preliminary prediction, the VF differences in the PSS and TBW both follow differences in temporal resolution: The PSS score was larger and the TBW width was wider in the central VF than in the peripheral VF.”

Point 12: 532

"A TOJ task"

Suggest: TOJ tasks are assumed to be...

Reply: I have revised the description as per your suggestion (line 541).

Point 13: 535

"a TOJ task"

Suggest: TOJ tasks are proposed

Reply: I have revised the description as per your suggestion (line 544).

Point 14: 562-563

Please clarify. I don't see how this relates to the previous sentence

Reply: I have deleted this sentence as per Reviewer 2’s comment.

Point 15: 574-575

Suggest: "Therefore, rapid recalibration may be induced by ...."

Reply: I have revised the description as per your suggestion (line 577). 

Replies to Reviewer 2’s comments

I appreciate your helpful and valuable comments on this manuscript and have revised it based on the comments.

1. In my previous point 1, I suggested that the author should introduce the critical differences of the mechanisms underlying the three paradigms used here. This is critical to help the author to predict possible results, and to provide readers clearer scope and motivation for the current study.

Reply: I have added descriptions regarding the differences in the mechanisms underlying the three paradigms (lines 92–95, lines 103–105).

“The underlying mechanisms differ between the TOJ and SJ tasks. The TOJ task reflects temporal discrimination processes, whereas the SJ task reflects temporal binding processes [26]. Furthermore, the differences between the TOJ and SJ tasks stem from their decisional and response processes [27].”

“Apparent causality among visual and auditory events as SB perception affects audio-visual synchrony perception in early multisensory integration processes [31].”

2. In my previous point 2, I have proposed that the dissociation between processing time and temporal resolution, which may lead to different predictions for PSS and TBW, respectively. Now the author suggested that the visual latency (which is close to the idea of processing time) is shorter in the center than in the periphery, which can explain the PSS result in the TOJ task. There should be hypothesis based on visual latency as a function of visual field addressed in Introduction.

Reply: I have added a hypothesis based on visual latency as a function of visual field (lines 74–84).

“The retinal positions of vision have a similar difference in these two channels: The central vision has a low temporal resolution, while the peripheral vision has a high temporal resolution [23]. Furthermore, there is a difference in visual latency between the central and peripheral visual fields (VFs): The response time for the visual stimulus is shorter at the central VF than it is at the peripheral VF [24]. Hence, the processing speed differs between the central and peripheral VFs. The present study confirmed the effects of visual temporal resolution on audio-visual synchrony perception by manipulating the VF in which the visual stimulus is presented (i.e., eccentricity). If synchrony perception for audio-visual stimuli follows differences in temporal resolution, then the TBW of the central VF would be wider than that of the peripheral VF, whereas if it follows the difference in visual latency, then the PSS of the central VF would be lower than that of the peripheral VF.”

3. P. 12, Keane et al. (2020) demonstrated the “choice-repetition bias” account based on reanalyzing Roseboom’s (2019) data; Keane et al., instead, reported a null rapid recalibration effect in Experiment 1, and a typical rapid recalibration effect in Experiment 2. Hence, the author’s “blurry visual information” account cannot work. Following the argument of “choice-repetition bias”, is it possible that the more pronounced rapid recalibration effect in both visual fields was a result of over-estimation due to this bias?

Reply: Keane et al. (2020) have shown in Experiment 2 that suppressing the choice-repetition bias leads to the typical rapid recalibration effect in the TOJ task. Therefore, in the TOJ task, the choice-repetition bias would be suppressed in the central VF. The choice-repetition bias is associated with the information reliability. Therefore, I have revised the interpretation on the basis of information reliability (lines 274–278).

“In this study, choice-repetition bias was suppressed in the central VF. Low information reliability induces a larger choice-repetition bias [39, 40]. Therefore, the reliability of judging temporal order for audio-visual stimuli would be high in the central VF. This speculation needs to be further investigated.”

4. Given that the motion display was longer than 1 sec, was the participant’s eye movement monitored or controlled?

Reply: The participants were instructed to gaze at the fixation cross during the motion display period. However, eye movement was not monitored. I have added a description of this point (line 430, lines 502–506).

“In the motion display period, participants were instructed to gaze the fixation cross.”

“Audio-visual temporal processing was not affected by VF differences in SB perception. SB perception did not differ in terms of the PSS and TBW between the central and peripheral VFs. However, the duration of motion display was over 1000 ms and participants’ eye movements were not monitored in this experiment. Thus, the participants’ gaze was off the fixation point in the motion display period, which may have affected the current results.”

5. In Experiment 3, given the fact that the PSS was not significantly different in the center and periphery (p = .69), I don’t see any explanation is needed. Specifically, the author’s suggestion that the participants made different judgments in the center and periphery is not convincing, and so is the following paragraph for the rapid recalibration effect.

Reply: I have deleted the statements regarding the VF differences in SB perception based on your comment.

6. Table 1: in the summary of rapid recalibration, it is weird to compare the results in the center and periphery either in the A-leading PSS or in the V-leading PSS. The summary should be the comparison between the A-leading PSS and in the V-leading PSS in the center and periphery, respectively.

Reply: I have revised Table 1 as per your suggestion.

7. Here are some arguments that I cannot follow:

(1) P. 10, lines 204-205: I do not understand this argument since the results were analyzed and presented separately in the central and peripheral visual field.

(2) P. 25, lines 533-535: If eccentricity effect can be compensated in later visual processing, and the TOJ involved higher-order processing, how could the former explain the PSS difference in the center and periphery in the TOJ?

Reply: (1) I did not split the visual field of a previous trial (Trial t-1); the visual field of the current trial (Trial t) was split. I have clarified the description of this point. (lines 216–218).

“The VFs in the previous trial (Trial t-1) were not split because the correspondence of the spatial location between the current (Trial t) and previous (Trial t-1) trials did not affect the rapid recalibration [37].”

(2) Later visual processing compensated the visual field difference of temporal resolution. It is not clear whether the visual field differences in visual latency are compensated in a later visual processing. The visual field difference of the PSS was consistent with that of visual latency in Experiment 1. Therefore, I interpreted that visual latency was not compensated in higher-order visual processing, and that synchrony perception was consistent with the visual field difference of visual latency in the TOJ task.

8. There are still some mistakes in the manuscript:

(1) P. 8, line 170: In Experiment 1, there were 10 SOAs. If there were 80 trials in a block, then there should be only 8 trials for each SOA. Same problem on p. 14, line 306, and p. 19, line 422.

(2) P. 11, line 223, the PSS should be “lower” or “smaller” in the center than in the periphery, since the values were negative.

(3) P. 14, line 298, and p. 19, line 412: it is unclear to me why the author insists to keep the “±0 ms” SOA, since it does not make sense at all.

(4) In the Procedure in Experiment 3, was there also double number of trials at the 0 ms SOA than other SOAs?

(5) P. 28, line 603: “…rapid temporal recalibration occurred only at the periphery VF in the TOJ…” should be at the central VF.

Reply: (1) I have revised the descriptions (line 182, 316, and 433).

(2) I have revised the term (line 235).

(3) I have revised the statement as per your suggestion (lines 316–317 and 433–434).

(4) You are right. I have revised the statement (line 308 and 422).

(5) I have revised the description (line 606).

---

## [Decision Letter · Decision Letter 2]

22 Oct 2021

PONE-D-20-40727R2Visual field differences in temporal synchrony processing for audio-visual stimuliPLOS ONE

Dear Dr. Takeshima,

Thank you for submitting your manuscript to PLOS ONE. After careful consideration, we feel that it has merit but does not fully meet PLOS ONE’s publication criteria as it currently stands. Therefore, we invite you to submit a revised version of the manuscript that addresses the points raised during the review process.

We look forward to receiving your revised manuscript.

Kind regards,

Deborah Apthorp, Ph.D

Academic Editor

PLOS ONE

Journal Requirements:

Additional Editor Comments (if provided):

The reviewers still have several concerns with the paper, in particular that the arguments are not very coherent. However, I feel the paper could be improved sufficiently to be publishable in PLoS one. I am unwilling to send this paper back to the reviewers as it has already gone through two rounds of revision. Thus, if the authors could carefully address each of the reviewers' comments (excluding Reviewer 1's suggestion to turn the paper into 3 separate papers), I will assess the manuscript myself and make a decision regarding publication.

Reviewers' comments:

Reviewer's Responses to Questions

**Comments to the Author**

1. If the authors have adequately addressed your comments raised in a previous round of review and you feel that this manuscript is now acceptable for publication, you may indicate that here to bypass the “Comments to the Author” section, enter your conflict of interest statement in the “Confidential to Editor” section, and submit your "Accept" recommendation.

Reviewer #1: All comments have been addressed

Reviewer #2: (No Response)

2. Is the manuscript technically sound, and do the data support the conclusions?

Reviewer #1: Yes

Reviewer #2: Partly

3. Has the statistical analysis been performed appropriately and rigorously? 

Reviewer #1: Yes

Reviewer #2: Yes

4. Have the authors made all data underlying the findings in their manuscript fully available?

Reviewer #1: Yes

Reviewer #2: (No Response)

5. Is the manuscript presented in an intelligible fashion and written in standard English?

Reviewer #1: Yes

Reviewer #2: No

6. Review Comments to the Author

Reviewer #1: The manuscript is improved and most of my comments have been addressed. It still reads as a collection of disparate results and tasks. Whilst this doesn’t make for a satisfying story it is arguably an accurate reflection of the state of the current empirical literature on AV synchrony perception. For this reason, the paper makes a useful contribution. I encourage the author to make this point in the Discussion.

Minor point

Line 81-82

I don't follow the presented logic of why AV temporal bandwidth is predicted to be broader in the central than in the peripheral visual field.

Reviewer #2: This is my third time reading this manuscript. Unfortunately, I still think that the current version of manuscript does not provide a clear rationale and comprehensive view of the study – it remains in a superficial state. I also think that the author simply included my words in previous comments in the current manuscript, rather than providing coherent explanations or elaborating these points.

The main problem is that the author included too many issues in a paper that are: 1) the PSS, width, and rapid recalibration of temporal binding window; 2) their differences in the central and peripheral visual fields; and 3) in three experimental paradigms. Not to mention that the author also had to address the inconsistent results in previous studies, as well as the novel results in the current study. My suggestion is that the author should separate three experiments into different papers, so the author can well discuss the results or conduct further experiments. Here are my concerns of the manuscript:

1. The main purpose of the current study, I assume, is stated in the title “visual field differences in temporal processing for audio-visual stimuli”. Given the fact that the author obtained different results in the three experimental paradigms, it is hard for me to judge which result pattern is genuinely attributed to the influence of eccentricity at the level of perceptual processing. Even if the author accepted my previous suggestions that TOJ involves post-perceptual processing compared to SJ, and SB involves causal relations between the audio-visual stimuli, explanations regarding the relationship between the results at difference filed and these mechanisms remains lacking.

For example, in TOJ, the PSS was more negative (i.e., at the auditory-leading side) in the center than in the periphery. On p. 13, the author stated that this difference was consistent with “difference in visual latency between the central and peripheral VFs”. But later the author said “could be attributed to differences in the response time”. Was this a perceptual effect or decisional effect? Would the patter be reversed if the task is changed into “which stimuli comes second”?

Another example on the top of p. 7: I don’t see the link between “the rapid temporal recalibration in the central and peripheral visual fields” and “the reversed pattern of PSS shift in TOJ and SJ reported by Roseboom”.

One more example is that, on p. 7, if the author’s previous study did not observe different rapid temporal recalibration based on different spatial frequency, then what is the reason to assume that there would be difference in the center vs. peripheral visual fields?

2. The author accepted previous suggestions and argued that temporal resolution is related to TBW whereas visual latency is related to PSS (p. 5). However, on p. 13, the authors still contrasted temporal resolution and visual latency as two possible explanations for PSS. The same confusion appeared again on p. 26.

3. Some sentences are still hard to follow or incorrect, such as:

p. 4: “Moreover, rapid recalibration, in which a prolonged period of adaptation is not necessary to observe temporal recalibration, has been reported.” – I don’t get it.

p. 8: the sentence regarding the power analysis is too long and many “and”.

p. 15 and p. 20: there were only 9 SOAs in SJ and SB tasks.

p. 19: “The binding for visual and auditory stimuli must be more flexible at the central VF, given the low temporal resolution” I don’t see the relations.

p. 22, line 456: should be Figs 7b and 7c

p. 24 “If SB judgment does not cause a change in the PSS due to the timing of sound in a previous trial” How come a participant’s judgment can cause something?

7. PLOS authors have the option to publish the peer review history of their article (what does this mean?). If published, this will include your full peer review and any attached files.

Reviewer #1: No

Reviewer #2: No

---

## [Author Response · Author response to Decision Letter 2]

14 Nov 2021

Replies to Reviewer 1’s comments

I appreciate your helpful and valuable comments on this manuscript. I have added the discussion related to the inconsistency of results among three methods.

Main point

The manuscript is improved and most of my comments have been addressed. It still reads as a collection of disparate results and tasks. Whilst this doesn’t make for a satisfying story it is arguably an accurate reflection of the state of the current empirical literature on AV synchrony perception. For this reason, the paper makes a useful contribution. I encourage the author to make this point in the Discussion.

Reply: I have discussed the disparate results among three method in General discussion section (lines 595–601).

“The PSS score tends to be a negative value (i.e., interpreted as synchrony when auditory stimulus leads to the visual stimulus) in a TOJ task compared to an SJ task [28]. SB perception is also likely to show a negative PSS value compared to an SJ task, since causality judgment is performed in SB perception [47]. In this study, the PSS scores indicated a similar tendency to these functional characteristics. One of the potential contributions is that each of these three methods measured different functional characteristics for audio-visual synchrony perception in a series of experiments.”

Minor point

Line 81-82

I don't follow the presented logic of why AV temporal bandwidth is predicted to be broader in the central than in the peripheral visual field.

Reply: I predicted that temporal bandwidth becomes broader to make the AV synchrony perception tolerant in the central VF, because the timing perception of visual presentation is ambiguous in the central VF due to its low temporal resolution. I have revised the description to clarify this point (lines 80–83).

“If synchrony perception for audio-visual stimuli follows differences in temporal resolution, then the TBW of the central VF would be wider than that of the peripheral VF for tolerant synchrony perception, because the timing perception of visual presentation is ambiguous in the central VF due to low temporal resolution.” 

Replies to Reviewer 2’s comments

I appreciate your helpful and valuable comments on this manuscript and have revised it based on the comments. However, I disagree that about separating the three experiments into different papers, because I considered that one of the useful contributions of this paper is to report the inconsistency of results among the three methods.

1-1. The main purpose of the current study, I assume, is stated in the title “visual field differences in temporal processing for audio-visual stimuli”. Given the fact that the author obtained different results in the three experimental paradigms, it is hard for me to judge which result pattern is genuinely attributed to the influence of eccentricity at the level of perceptual processing. Even if the author accepted my previous suggestions that TOJ involves post-perceptual processing compared to SJ, and SB involves causal relations between the audio-visual stimuli, explanations regarding the relationship between the results at difference filed and these mechanisms remains lacking.

Reply: I believe that the explanation regarding the VF differences shown in the present study is sufficient. However, the statements were partly unclear in the previous manuscript, and I have, therefore, revised these points.

1-2. For example, in TOJ, the PSS was more negative (i.e., at the auditory-leading side) in the center than in the periphery. On p. 13, the author stated that this difference was consistent with “difference in visual latency between the central and peripheral VFs”. But later the author said “could be attributed to differences in the response time”. Was this a perceptual effect or decisional effect? Would the patter be reversed if the task is changed into “which stimuli comes second”?

Reply: I consider that the difference in PSS between the central and peripheral VFs is the perception effect. Breitmeyer (1985) reported that response time reflects differences in visual latency. Thus, the difference in response time between central and peripheral VFs would also be a perceptual effect caused by difference in visual latency. I have added the statements of this point (lines 259–266).

“The difference in the PSS observed between the central and peripheral VFs was consistent with the difference in visual latency between the central and peripheral VFs. The response time for the visual stimulus was shorter in the central VF than in the peripheral VF [24]. Previous studies that manipulated spatial frequency also observed differences in the PSS consistent with the differences observed in response time [19, 29]. The difference in visual latency is reflected in the response time of the visual stimulus [22]. Therefore, in a TOJ task, differences in the PSS between the central and peripheral VFs could be attributed to differences in visual latency rather than temporal resolution.”

1-3. Another example on the top of p. 7: I don’t see the link between “the rapid temporal recalibration in the central and peripheral visual fields” and “the reversed pattern of PSS shift in TOJ and SJ reported by Roseboom”.

Reply: There was a mistake in the description; therefore the correct wording has been revised (lines 110–116).

“In addition, the differences in the effects of rapid temporal recalibration between SJ and TOJ tasks were investigated to confirm the effects of VFs on audio-visual temporal processing in this study. Roseboom [32] showed the opposite change in PSS due to rapid temporal recalibration between SJ and TOJ tasks: the PSS changed in the same direction as the stimulus onset asynchrony (SOA) of previous trials in the SJ task and in the opposite direction in the TOJ task. Therefore, this difference in the rapid temporal recalibration between SJ and TOJ tasks was also investigated.”

1-4. One more example is that, on p. 7, if the author’s previous study did not observe different rapid temporal recalibration based on different spatial frequency, then what is the reason to assume that there would be difference in the center vs. peripheral visual fields?

Reply: I considered that the VF difference in PSS change between central and peripheral VFs is related to the magnitude of the choice-repetition bias in the TOJ task. In this task, unlike the SJ task, the choice-repetition bias operates the process of PSS change. Thus, it is possible that the magnitude of the choice-repetition bias caused the VF difference in the PSS change, not the rapid recalibration process. I have clarified this explanation in the revised manuscript (lines 267–277).

“The results of normal rapid recalibration for the central VF were inconsistent with the predictions based on previous studies. Roseboom [32] showed that the PSSs shifted in opposite directions from normal rapid temporal recalibration in a TOJ task. Moreover, Keane, Bland, Matthews, Carroll, and Wallis [38] found that opposite-directed PSS shifts were induced by choice-repetition bias in a TOJ task. Choice-repetition bias refers to the tendency to repeat judgments of the temporal order of a previous trial in a current trial. Additionally, rapid recalibration was obfuscated by opposite-directed PSS shifts to a choice-repetition bias [38]. It is possible that the choice-repetition bias was suppressed in the central VF in this study. Low information reliability induces a larger choice-repetition bias [39, 40]. Therefore, the reliability of judging temporal order for audio-visual stimuli would be high in the central VF. This speculation needs to be further investigated.”

2. The author accepted previous suggestions and argued that temporal resolution is related to TBW whereas visual latency is related to PSS (p. 5). However, on p. 13, the authors still contrasted temporal resolution and visual latency as two possible explanations for PSS. The same confusion appeared again on p. 26.

Reply: Thank you, I accepted your useful suggestion. However, it is also necessary to explain why visual latency, rather than temporal resolution, affected the PSS score in the TOJ task. I revised the descriptions to avoid confusion and clarify my arguments. (lines 259–266, lines 537–545).

“The difference in the PSS observed between the central and peripheral VFs was consistent with the difference in visual latency between the central and peripheral VFs. The response time for the visual stimulus was shorter in the central VF than in the peripheral VF [24]. Previous studies that manipulated spatial frequency also observed differences in the PSS consistent with the differences observed in response time [19, 29]. The difference in visual latency is reflected in the response time of the visual stimulus [22]. Therefore, in a TOJ task, differences in the PSS between the central and peripheral VFs could be attributed to differences in visual latency rather than temporal resolution.”

“As a preliminary prediction, the VF differences in the PSS and TBW both follow differences in each visual latency and temporal resolution: the PSS score was lower and the TBW width was wider in the central VF than in the peripheral VF. It has been shown that the difference in eccentricity-dependent temporal resolution is observed in the early visual cortex and is compensated later in the cortical visual pathway [42]. TOJ tasks are assumed to be associated with higher-order processing compared to SJ tasks [28, 43]. Therefore, what PSS score was affected by visual latency could be attributed to the visual hierarchy of eccentricity-dependent temporal contrast in a TOJ task.”

3. Some sentences are still hard to follow or incorrect, such as:

(1) p. 4: “Moreover, rapid recalibration, in which a prolonged period of adaptation is not necessary to observe temporal recalibration, has been reported.” – I don’t get it.

(2) p. 8: the sentence regarding the power analysis is too long and many “and”.

(3) p. 15 and p. 20: there were only 9 SOAs in SJ and SB tasks.

(4) p. 19: “The binding for visual and auditory stimuli must be more flexible at the central VF, given the low temporal resolution” I don’t see the relations.

(5) p. 22, line 456: should be Figs 7b and 7c

(6) p. 24 “If SB judgment does not cause a change in the PSS due to the timing of sound in a previous trial” How come a participant’s judgment can cause something?

Reply: (1) I revised the description to clarify my intention (lines 58–59).

“Moreover, van der Burg, Alais, and Cass [17] showed temporal recalibration without adaptation period, which they termed “rapid recalibration.””

(2) I revised the sentence as per your comment (lines 138–142).

“To determine the sample size needed for this study, PANGEA (https://jakewestfall.shinyapps.io/pangea/) was used to calculate the power (1-β) needed to detect a two-way interaction with following parameters: effect size (d) = 0.45, variance of error = 0.333, variance of two-way interaction = 0.083, and the number of condition repetitions = 24. PANGEA indicated a power of 0.86 when a sample size was 16 participants.”

(3) I corrected the statement (line 306 and 420).

(4) I revised description to clarify my argument (lines 383–385).

“Thus, the range of audio-visual synchrony perception would be more tolerant in the central VF due to adapting to low temporal resolution.”

(5) I corrected the description (line 455).

(6) I revised statements to clarify my argument (lines 508–513).

“It is possible that participants performed timing judgment of bounce in the central VF and causal judgment of bounce in the peripheral VF. The occurrence of rapid recalibration due to SB perception has not yet been examined. If participants performed a causal judgment in peripheral VF and the timing information on a previous trial did not cause a change in the PSS in causal judgment, it is consistent with the results of this experiment.”

---

## [Editor Report · Decision Letter 3]

25 Nov 2021

Visual field differences in temporal synchrony processing for audio-visual stimuli

PONE-D-20-40727R3

Dear Dr. Takeshima,

We’re pleased to inform you that your manuscript has been judged scientifically suitable for publication and will be formally accepted for publication once it meets all outstanding technical requirements.

Kind regards,

Deborah Apthorp, Ph.D

Academic Editor

PLOS ONE
---

## [Editor Report · Acceptance letter]

6 Dec 2021

PONE-D-20-40727R3 

Visual field differences in temporal synchrony processing for audio-visual stimuli 

Dear Dr. Takeshima:

I'm pleased to inform you that your manuscript has been deemed suitable for publication in PLOS ONE. Congratulations! Your manuscript is now with our production department. 

Kind regards, 

on behalf of

Dr. Deborah Apthorp 

Academic Editor

PLOS ONE